

# Sulfate Geoengineering Impact on Methane Transport and Lifetime: Results from the Geoengineering Model Intercomparison Project (GeoMIP)

Daniele Visioni[1,2], Giovanni Pitari[1], Valentina Aquila[3], Simone Tilmes[4], Irene Cionni[5], Glauco Di Genova[2], and Eva Mancini[1,2]

[1]Department of Physical and Chemical Sciences, Universitá dell'Aquila, 67100 L'Aquila, Italy
[2]CETEMPS, Universitá dell'Aquila, 67100 L'Aquila, Italy
[3]GESTAR/Johns Hopkins University, Department of Earth and Planetary Science, 3400 N Charles Street, Baltimore, MD 21218, USA
[4]National Center for Atmospheric Research, Boulder, CO 80305, USA
[5]ENEA, Ente per le Nuove Tecnologie, l'Energia e l'Ambiente, 00123 Roma, Italy

*Correspondence to:* Daniele Visioni (daniele.visioni@aquila.infn.it)

**Abstract.** Sulfate geoengineering, made by sustained injection of $SO_2$ in the tropical lower stratosphere, may impact the $CH_4$ abundance through several photochemical mechanisms affecting tropospheric OH and hence the methane lifetime. (a) Solar radiation scattering increases the planetary albedo and cools the surface, with a tropospheric $H_2O$ decrease. (b) The tropospheric UV budget is upset by the additional aerosol scattering and stratospheric ozone changes: the net effect is meridionally not uniform, with a net decrease in the tropics, thus producing less tropospheric $O(^1D)$. (c) The extratropical downwelling motion from the lower stratosphere tends to increase the sulfate aerosol surface area density available for heterogeneous chemical reactions in the mid-upper troposphere, thus reducing the amount of $NO_x$ and $O_3$ production. (d) The tropical lower stratosphere is warmed by solar and planetary radiation absorption by the aerosols. The heating rate perturbation is highly latitude dependent, producing a stronger meridional component of the Brewer-Dobson circulation. The net effect on tropospheric OH due to the enhanced stratosphere-troposphere exchange may be positive or negative depending on the net result of different superimposed species perturbations ($CH_4$, $NO_y$, $O_3$, $SO_4$) in the extratropical upper troposphere and lower stratosphere (UTLS). In addition, the atmospheric stabilization resulting from the tropospheric cooling and lower stratospheric warming favors an additional decrease of the UTLS extratropical $CH_4$, by lowering the horizontal eddy mixing. Two climate-chemistry coupled models are used to explore the above radiative, chemical and dynamical mechanisms affecting $CH_4$ transport and lifetime (ULAQ-CCM and GEOSCCM). The $CH_4$ lifetime may become significantly longer (by approximately 16%) with a sustained injection of 8 Tg-$SO_2$/yr started in year 2020, which implies an increase of tropospheric $CH_4$ (200 ppbv) and a positive indirect radiative forcing of sulfate geoengineering due to $CH_4$ changes (+0.10 W/m$^2$ in the 2040-2049 decade and +0.15 W/m$^2$ in the 2060-2069 decade).



# 1 Introduction

Many geoengineering methods have been proposed in order to temporarily balance out the direct effect of the increase of anthropogenic greenhouse gases emissions (Kravitz et al. (2011)). Amongst those, stemming from the observations of the effects of large volcanic eruptions, is the injection of sulfate aerosol precursors (e.g, $SO_2$) into the stratosphere (Crutzen (2006),

Robock et al. (2011), Kravitz et al. (2012)). The injection above the tropopause of very large amounts of particles and sulfur gases due to explosive volcanic eruptions is able to increase the stratospheric aerosol optical depth by more than one order of magnitude. The initial volcanic $SO_2$ plume quickly nucleates into $H_2SO_4$ vapor (Bluth et al. (1992)) producing an optically thick cloud of sulfate aerosols (McCormick and Veiga (1992), Lambert et al. (1993), Long and Stowe (1994)). The high reflectivity of these aerosols at visible and UV wavelengths effectively decreases the amount of solar radiation reaching the Earth

surface, thus producing a net global cooling. In 1991, for example, the Pinatubo eruption produced a reduction in the global surface air temperature estimated to be a value ranging from 0.5 K (Soden et al. (2002)) down to 0.14 K globally if recent, detrended analyses (Canty et al. (2013)) are considered.

Beside the direct effect on surface temperatures, however, there is the need for a thorough examination of other effects on

atmospheric circulation and chemical composition of the troposphere and stratosphere brought about by the increase in lower stratosphere optical thickness (Visioni et al. (2017)). First of all, connected with the increased radiation scattering by the $H_2SO_4$ particles comes an increase in the lower stratospheric diabatic heating rates, caused by the direct aerosol absorption in the near-infrared wavelengths. This causes a local positive temperature change (Labitzke and McCormick (1992)) which induces a significant increase of westerly winds from the thermal wind equation, with peaks at mid-latitudes in the mid-stratosphere

(Pitari et al. (2016c)). These dynamical changes tend to increase the amplitude of planetary waves in the stratosphere and to enhance the tropical upwelling in the rising branch of the Brewer Dobson circulation (Pitari et al. (2014), Aquila et al. (2014)). For continuity, a stronger downward component is found in the lower branch of the Brewer-Dobson circulation (Aquila et al. (2013), Pitari et al. (2016b)).

These dynamical changes can bring about modification in the concentration and growth-rate of long-lived species that act as greenhouse gases, such as $N_2O$ and $CH_4$, as observed in the case of the Pinatubo eruption (Schauffler and Daniel (1994), Dlugokencky et al. (1994)): a heightened exchange between the stratosphere and the troposphere, with an increase in the downward mid and high latitude fluxes would mean an injection of stratospheric air containing smaller mixing ratios of such gases in the troposphere. In addition, the horizontal eddy mixing of UTLS tropical mixing ratios with the extra-tropics is lowered

as a consequence of the atmospheric stabilization resulting from the tropospheric cooling and lower stratospheric warming: this favors an additional decrease of the UTLS extratropical downward fluxes of $CH_4$ and other long-lived species (Pitari et al. (2016b)). The overall effect on tropospheric OH due this enhanced stratosphere-troposphere exchange and perturbed UTLS horizontal mixing may be positive or negative depending on the net result of different superimposed species perturbations in





the UTLS ($CH_4$, $NO_y$, $O_3$).

Coupled with this perturbation of the stratosphere-troposphere exchange, the lifetime of long-lived species with tropospheric OH sink can also be modified by other changes brought about by an injection of tropical stratospheric aerosols: a) the surface cooling would directly lessen the amount of water vapor, thus lowering the tropospheric OH concentration; b) the tropical tropospheric UV decrease due to enhanced radiation scattering would reduce the production of $O(^1D)$, thus decreasing OH production from $O(^1D) + H_2O$; c) the increasing aerosol surface area density (SAD) would enhance heterogeneous chemistry in the mid-upper troposphere, which reduces the amount of $NO_x$ and the rate of $O_3$ production, both negatively affecting the amount of tropospheric OH. Since $CH_4$ is depleted by the OH radical, all these changes would mean an increase in methane lifetime (Banda et al. (2013), Banda et al. (2015)). The aim of this study is to evaluate the chemical, radiative and dynamical effects of a sustained injection of $SO_2$ in the stratosphere on the lifetime and abundance of $CH_4$. The characteristics of the experiment follow the description of experiment G4 in the Geoengineering Model Intercomparison Project (GeoMIP) (Kravitz et al. (2011)). For this experiment, the background anthropogenic forcing profile corresponds to the one from the Representative Concentration Pathway 4.5 (RCP4.5) (Taylor et al. (2012)). Starting from 2020, 8 (or 5) Tg-$SO_2$/yr are injected in the stratosphere with a sudden stop after 50 years. Additional 20 years of model simulations are performed (up to 2090) in order to assess the termination effects of the sulfur injection.

The paper is organized in seven subsequent parts. Section 2 includes a description of participating models. In Section 3 a model evaluation for long lived species stratospheric abundance and transport is presented using available satellite observations. Section 4 analyses the sulfate geoengineering induced perturbations on stratospheric species transport, while Section 5 discusses the effects on tropospheric chemistry and $CH_4$ direct and indirect radiative forcing components, with the overall main conclusions discussed in Section 6.

## 2   Model experiments

The main features of the participating models are summarized in Table 1. One of these models (CCSM-CAM4) is an atmosphere-ocean coupled model and it has been used (without interactive chemistry) to calculate the surface temperature evolution from 2010 to 2090 for a reference RCP4.5 case and a geoengineering G4 perturbed case with 8 Tg-$SO_2$/yr injected continuously from 2020 to 2070 (Kravitz et al. (2011); Pitari et al. (2014)). One of the other two models (ULAQ-CCM) has assimilated the sea surface temperatures (SST) calculated in the CCSM-CAM4 atmosphere-ocean coupled model for the reference RCP4.5 and the perturbed G4 cases (i.e., two different SST datasets, both without interactive chemistry), whereas the third model (GEOSCCM) has run the G4 case with RCP4.5 SSTs assimilated from the CESM atmosphere-ocean coupled model. A more detailed description of these numerical models can be found in Tilmes et al. (2016) and Pitari et al. (2014).





In order to properly assess the different contributions to $CH_4$ changes discussed before, three different experiments have been carried out with the ULAQ-CCM model: experiments (a,b) use appropriate SSTs for RCP4.5 and G4 cases (as previously explained), with MBC and FBC approaches for (a) and (b), respectively. Experiment (c), on the other hand, uses the same SST for both RCP4.5 and G4 cases (as in GEOSCCM), with the purpose of highlighting the impact of SST changes on the G4-RCP4.5 large scale transport perturbations. The full list of numerical experiments completed with the three models is presented in Table 2.

**Table 1.** Summary of main model features. Column 6 includes the stratospheric aerosol effective radius (reff in $\mu$m) at 20 km over the tropics (2040-2049). Values deduced from SAGE-II observations are: 0.22±0.02 $\mu$m as an average over 1999-2000 for unperturbed background conditions and 0.57±0.03 $\mu$m as an average over July 1992-June 1993 for a volcanic perturbation (i.e., Pinatubo) comparable in magnitude to G4 with 5 Tg-SO2 injection (in terms of average stratospheric mass burden of sulfate). G4 aerosols are injected at the equator between 16 km and 25 km altitude. MBC → Mixing ratio Boundary Condition. FBC → Flux Boundary Condition.

| Model | Resolution[1] | Ocean | QBO | $CH_4$ Surface Boundary Condition | Stratospheric Aerosol Source |
|---|---|---|---|---|---|
| CCSM-CAM4 | 1.9° × 2.5°, L40 Top : 3 hPa | Coupled | No | MBC | From $SO_2$ oxidation[2] G4 → 8 Tg-$SO_2$ [Tilmes et al. (2016)] |
| GEOSCCM | 2° × 2.5°, L72 Top : 0.01hPa | Prescribed SSTs [CESM4] G4=RCP4.5 | Internal[3] | MBC | From $SO_2$ oxidation[2] G4 → 5 Tg-$SO_2$ G4 → $r_{eff}$ = 0.60 $\mu$m |
| ULAQ-CCM (a) | 5° × 6°, L126 Top : 0.04hPa | Prescribed SSTs [CCSM-CAM4] | Nudged | MBC | From $SO_2$ oxidation[4] G4 → 8 Tg-$SO_2$ G4 → $r_{eff}$ = 0.78 $\mu$m |
| ULAQ-CCM (b) | As above | As above | As above | FBC | As above |
| ULAQ-CCM (c) | As above | Prescribed SSTs [CCSM-CAM4] G4=RCP4.5 | As above | MBC | From $SO_2$ oxidation[4] G4 → 5 Tg-$SO_2$ G4 → $r_{eff}$ = 0.61 $\mu$m |

[1] Latitude by longitude horizontal resolution, number of vertical layers, and model top atmospheric pressure.

[2] forced with background aerosols from SAGE-II data for 1999.

[3] QBO internally generated using a gravity wave drag parameterization and resolved wave forcing.

[4] ULAQ-CCM includes aerosol microphysics (RCP4.5 $r_{eff}$ = 0.19 $\mu$m)





**Table 2.** Summary of numerical experiments. Ensemble size.

| Model | RCP4.5 | G4 | Used for |
|---|---|---|---|
| CCSM-CAM4 | 2 | 2 | SSTs for the ULAQ-CCM simulation |
| GEOSCCM | 3 | 3 | Assessing $CH_4$ changes due to transport |
| ULAQ-CCM (a) | 2 | 2 | Assessing $CH_4$ changes due to transport |
| ULAQ-CCM (b) | 2 | $2 + 1^1 + 1^2 + 1^3 +$ | Assessing $CH_4$ changes due to chemistry |
| ULAQ-CCM (c) | 2 | $2 + 1^4$ | Assessing $CH_4$ changes due to transport and chemistry |

[1] FBC sensitivity case [sn1] with temperature and winds from RCP4.5 in the chemistry module and continuity equations of chemical tracers.

[2] FBC sensitivity case [sn2] with temperature from RCP4.5 in the chemistry module.

[3] FBC sensitivity case [sn3] with winds from RCP4.5 in the continuity equations of chemical tracers.

[4] MBC sensitivity case with RCP4.5 SSTs, but 8 Tg-$SO_2$/yr injection.

The ULAQ-CCM sensitivity cases run with the FBC approach will help in assessing the role of temperature and wind changes in the $CH_4$ lifetime perturbation under geoengineering conditions.

# 3    Model evaluation

In order to properly evaluate the models, different sets of observations have been employed (Table 3). $CH_4$ measurements are taken by the Halogen Occultation Experiment (HALOE), which is on board of the Upper Atmosphere Research Satellite (UARS), launched in 1991 (Russell et al. (1993)). Climatologies are formed for the period 1991-2005, based on extended datas from Grooss and Russell III (2005). HALOE measurements range from 15 to 60-130 km altitude (depending on the species) and cover 80°S to 80°N in latitude within one year. In all intercomparisons the HALOE climatological mean and the interannual standard deviation ($1\sigma$) are shown. $CH_4$ and $N_2O$ profiles are estimated by Aura TES thermal infrared radiances at $\lambda$=8 $\mu$m with version 5 retrieval algorithm, where $CH_4$ is corrected using co-retrieved $N_2O$ estimates (Worden et al. (2012)). Climatological mean and inter annual standard deviation for both species are calculated for the period 2004-2010. Climatological mean and inter annual standard deviation of $N_2O$ between 2001-2005 are based on Odin/SMR product (Urban et al. (2009)).

**Table 3.** Summary of $CH_4$ and $N_2O$ satellite observations used in this study.

| Observation | $CH_4$ | $N_2O$ |
|---|---|---|
| TES | 2004-2010 | 2004-2010 |
| HALOE | 1991-2005 | |
| SMR | | 2001-2005 |

$CH_4$ diagnostics largely reflect the skill of the transport representation in the models. We examined climatological zonal profiles at selected latitudes, months, and pressure levels, for both model outputs and observations (Fig. 1). The climatologies refer to the years 1990-2010, in order to include the range of HALOE and TES observations. Both ULAQ-CCM and GEOSCCM



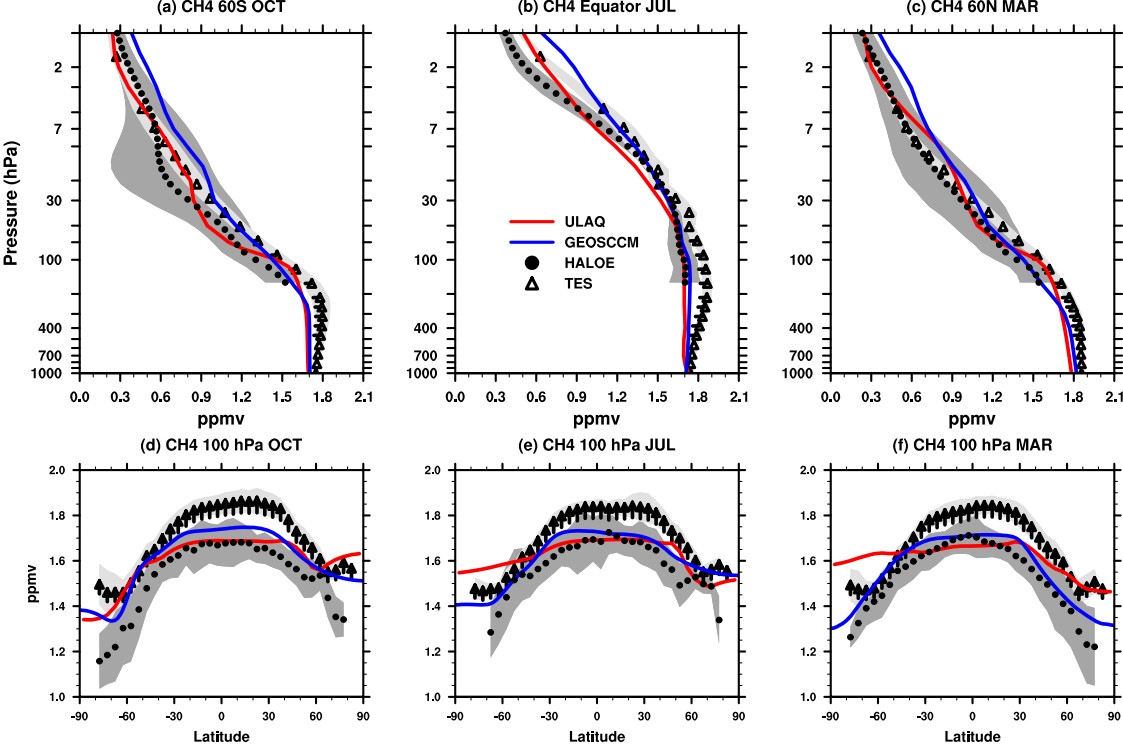

**Figure 1.** Evaluation of zonal and annual mean $CH_4$ mixing ratios ULAQ-CCM (red) and GEOSCCM (blue) simulations averaged over 1991-2010. Observations are taken from HALOE (black dots, average 1991-2005) (Grooss and Russell III (2005)) and TES Aura.

compare well with observations and are normally in the $\pm 1\sigma$ deviation interval, relative to the climatological zonal mean. Some spread between models appears, more evidently in the polar regions at 100 hPa. GEOSCCM values are generally closer to observations than those of ULAQ-CCM. Otherwise, models perform quite similarly, and overall these diagnostics do not reveal weakness in the simulations.

5    The correlation between $CH_4$ and $N_2O$ can be used to investigate transport properties relative to model and observations (SPARC-CCMVal (2010)). Fig. 2 and Fig. 3 show $CH_4$ vs. $N_2O$ correlations between 100 hPa and 1 hPa for models climatological mean in the same time range of TES (Fig. 2) and in the same time range of HALOE relative to $CH_4$ and SMR relative to $N_2O$ (Fig. 3). In Table 4a and Table 4b we present Pearson correlation coefficients relative to the different latitude bands in both cases. Confidence interval is calculated using the Fisher transform inverse. The existence of mixing barriers at the edge of

10   the tropical pipe allows the distinction between tropical (Fig. 2c, 2f, 3c, 3f) and midlatitude correlations (Fig. 2b, 2e, 3b 3f). All these panels show a compact correlation and a good agreement with the observations; the relative Pearson coefficients in Table 4a and Table 4b are always significative. Panels regarding polar regions (Fig. 2a, 2d and Fig. 3a, 3d) present a larger spread with slightly lower (but still significant) Pearson coefficient between 90S-60S. In the lower stratosphere at tropical and mid latitudes there is a strong compact relationship between $CH_4$ and $N_2O$ related to the slope equilibrium (Sankey and Shepherd (2003)):



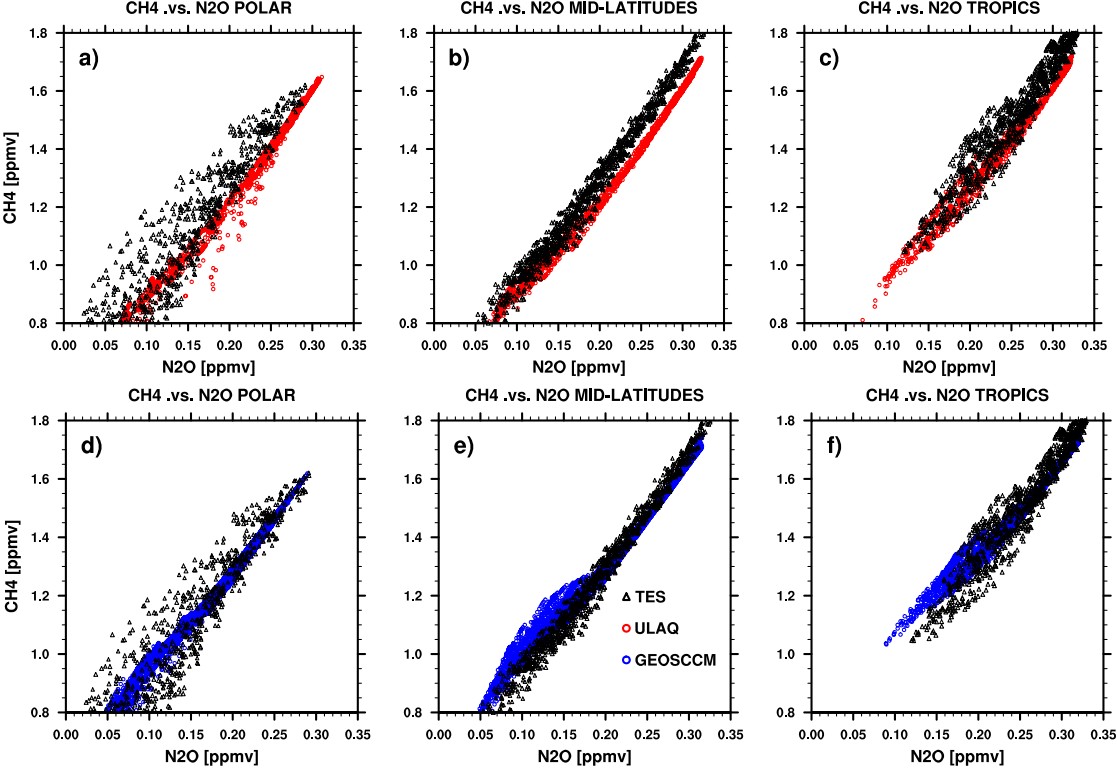

**Figure 2.** Scatter plots of zonal and monthly mean mixing ratio values of CH$_4$ and N$_2$O for ULAQ-CCM (red) and GEOSCCM (blue) simulations, in the layer 1-100 hPa and averaged over 2004-2010. The panels refer to latitude bands 60S-90S and 60N-90N (a), 30S-60S and 30N-60N (b), 30S-30N (c). Units are ppmv. Model values are evaluated with CH$_4$ and N$_2$O data from TES observations (black), averaged over 2004-2010.

the mixing happens at faster time scale than the chemical loss and transport to the surface. At polar latitudes the correlation is affected by vortex edge, which represents a mixing barrier during the winter-spring season. In polar regions, models display a correlation more compact compared with observed data: this happens because the latter are affected by a large uncertainty due to either: sparse coverage of the satellites data, as shown by Grooss and Russell III (2005) for HALOE or the low sensitivity

5   of the retrieval method as shown by Worden et al. (2012) for TES. Overall values in Table 4a present a better correlation with respect to values in Table 4b: this might be a consequence of a different range of years used for CH$_4$ (1991-2005) and N$_2$O (2001-2005).




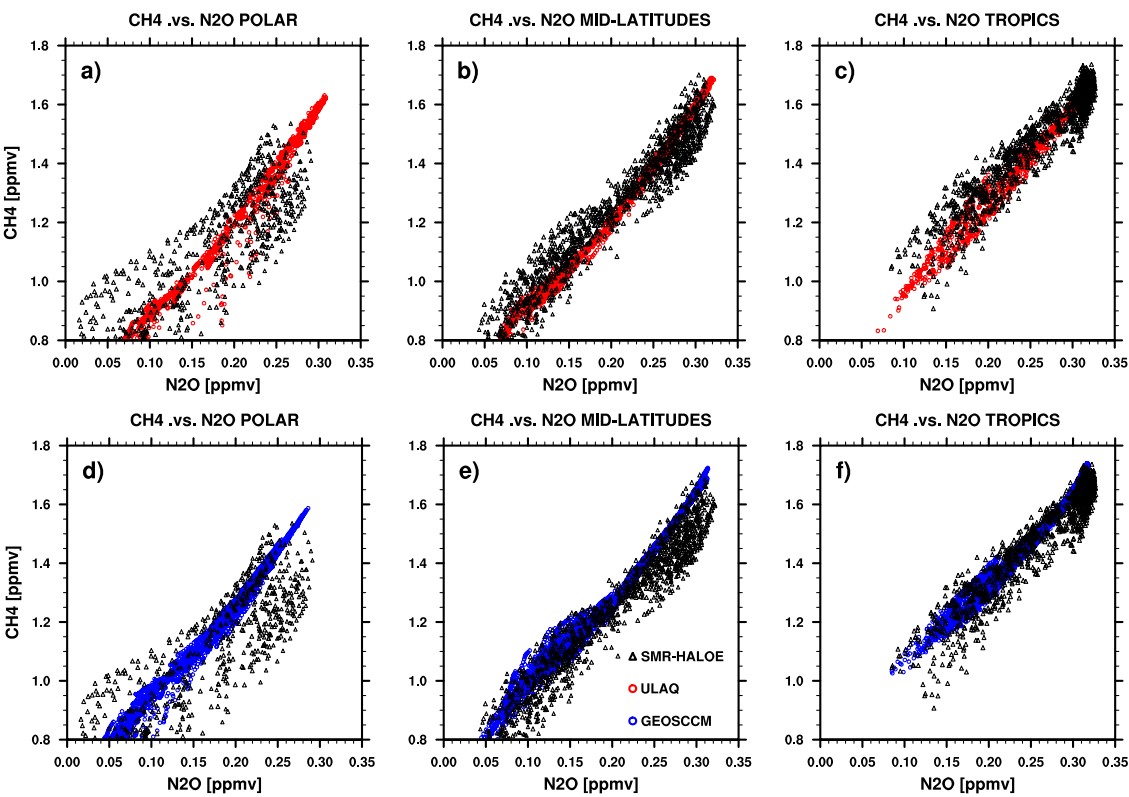

**Figure 3.** As in Fig. 2, but using observed data from HALOE for CH$_4$ (average 1991-2005) and SMR-Odin for N$_2$O (average 2001-2005) (Urban et al. (2009)). Model data are averaged over 1991-2005 for CH$_4$ and 2001-2005 for N$_2$O.





**Table 4.** a) Pearson correlation coefficient with associated confidence interval calculated using the Fischer transform inverse, for observations and model data presented in Fig. 2 (2004-2010). b) as in a) but for the data presented in Fig. 3 (1991-2005 for the models and HALOE, 2001-2005 for SMR).

| a) | | | | | |
|---|---|---|---|---|---|
| R_Pearson | 90S-60S | 60S-30S | 30S-30N | 30N-60N | 60N-90N |
| TES | 0.921 [0.908-0.933] | 0.988 [0.986-0.990] | 0.971 [0.967-0.974] | 0.995 [0.994-0.995] | 0.956 [0.948-0.962] |
| GEOSCCM | 0.982 [0.980-0.984] | 0.992 [0.990-0.992] | 0.990 [0.989-0.990] | 0.997 [0.997-0.997] | 0.994 [0.994-0.995] |
| ULAQ-CCM | 0.990 [0.988-0.991] | 0.996 [0.995-0.996] | 0.995 [0.994-0.995] | 0.997 [0.997-0.998] | 0.993 [0.992-0.994] |
| b) | | | | | |
| HALOE/SMR | 0.761 [0.723-0.794] | 0.958 [0.951-0.963] | 0.952 [0.947-0.957] | 0.970 [0.966-0.995] | 0.926 [0.914-0.938] |
| GEOSCCM | 0.978 [0.976-0.980] | 0.990 [0.989-0.991] | 0.990 [0.989-0.991] | 0.996 [0.996-0.997] | 0.995 [0.994-0.995] |
| ULAQ-CCM | 0.982 [0.979-0.985] | 0.995 [0.994-0.995] | 0.993 [0.992-0.993] | 0.996 [0.995-0.997] | 0.992 [0.991-0.993] |

Another important diagnostic for the evaluation of the model transport is based on the mean age of air (AoA). In particular, gradient between tropics and midlatitudes can be used to assess tropical ascend independently of quasi-horizontal mixing (SPARC-CCMVal (2010)). Fig. 4a and Fig. 4b show tropical and midlatitudes (35N-50N) profiles of mean AoA and Figure 4c displays vertical gradients of mean AoA between 45N and the equator. The mean AoA observations are based on Andrews

et al. (2001) and Engel et al. (2009) data, as presented by Strahan et al. (2011). Models profiles are very similar and in good agreement with observations. Following Strahan et al. (2011), tropical mean AoA profiles (Fig. 4a) combine the effect of ascent rate and horizontal mixing. The agreement of model and observations only shows that the combined effects of ascent and mixing produce a realistic mean AoA in the models. Fig. 4c identifies how ascent contribute to the overall tropical transport seen in Fig. 4a.

The horizontal gradient of mean age (Figure 4c) is able to reveal some characteristics of the Brewer-Dobson circulation (BDC) (Neu and Plumb (1999)), namely the ascent rate. In fact, differences between midlatitude and tropical values exclude horizontal mixing, since that affects equally both the tropics and midlatitudes. In GEOSCCM and ULAQ-CCM the horizontal gradient is smaller than observations up to 21 Km, indicating a fast ascent, but still included in the range of observed variability. The analysis of the relationship between mean AoA and $N_2O$ (Fig. 4d) validates the lower stratospheric transport and our use of

the well-measured $N_2O$ in Fig. 2 and Fig. 3. The models values of mean AoA and $N_2O$ shown represent the climatological mean (1980-2010) in the range 10-100 hPa and 10S-10N, while observed value of mean age of air are the same as in Fig.



4a and observed values of $N_2O$ are SMR/Odin climatological mean (2001-2005). The correlation for $N_2O > 150$ ppbv looks compact, the slope of the model curves is similar to the observed curve; models values of $N_2O$ and mean AoA are in the same range as the observations. Fig. 4e presents the evaluation of latitudinal sections of $N_2O$ at 50 hPa against SMR/Odin data. For tropical values, GEOSCCM and ULAQ-CCM agree very well with the observations, overall model values fall inside the $2\sigma$

5  interannual variability. At northern midlatitudes ULAQ-CCM overestimates SMR; in the Southern Hemisphere GEOSCCM values are larger than SMR and ULAQ-CCM values lower.

In order to properly asses the temperature of the polar stratosphere and its interannual variability, the models must correctly simulate the vertical propagation of planetary waves from the troposphere to the stratosphere. By knowing this, we can validate the models transport skill by looking at the correlation between polar temperature and a proxy for planetary wave propagation.

10  This can be done by looking at the correlation between the meridional heat flux at 100 hPa ($40°$ to $80°$ for the two hemispheres) and the 50 hPa ($60°$ to $90°$ for the two hemispheres) polar temperatures (Eyring et al. (2006)). This is shown in Table 5, where we compare the coefficient of the linear fit between the two quantities for ULAQ-CCM, GEOSCCM and the ERA40 reanalysis. The positive slope is found in both models and reanalysis, with a greater similarity in the Northern Hemisphere with respect to the Southern Hemisphere: this difference was already shown in Eyring et al. (2006).



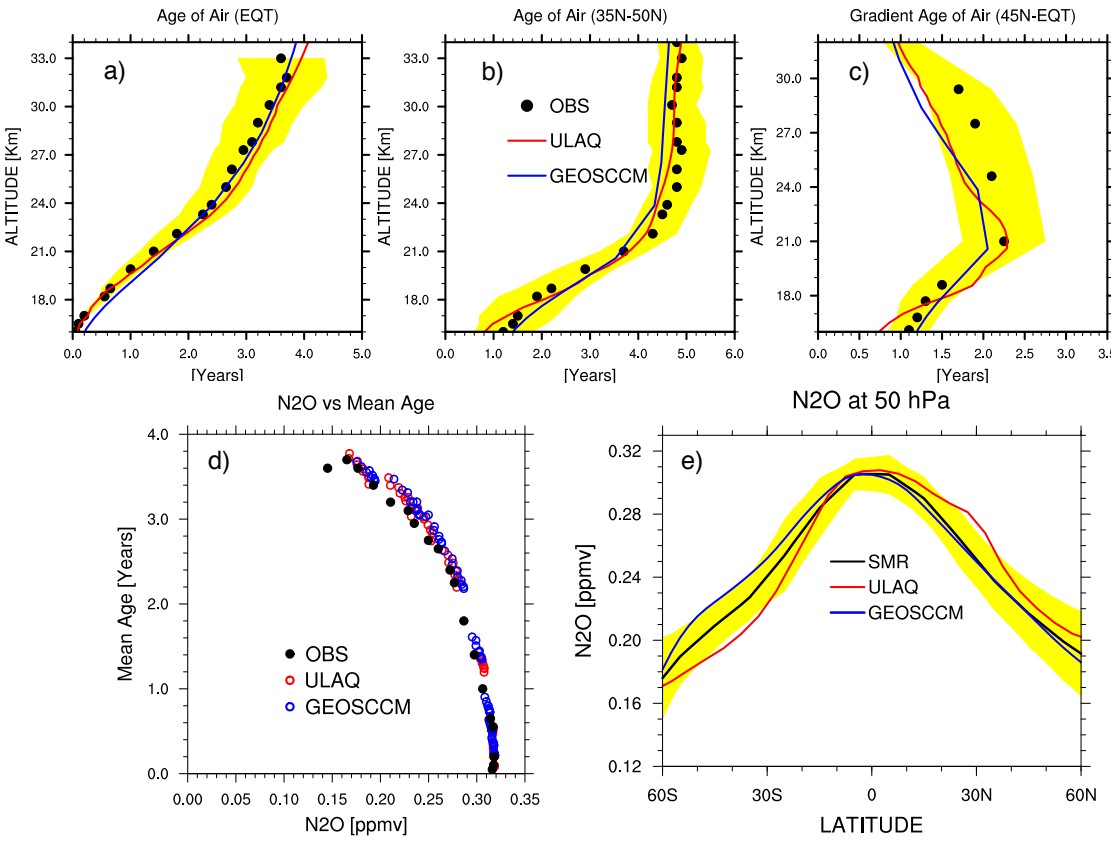

**Figure 4.** Vertical profiles of (a) equatorial and (b) mid-latitude AoA for GEOSCCM (blue line) and ULAQ-CCM (red line), compared with the range of observations from Andrews et al. (2001) and Engel et al. (2009) (yellow-filled area). The time average is from 1980 to 2000; the latitudinal average is 10S-10N in (a) and 35N-50N in (b). The latitudinal gradient of AoA is shown in panel (c), calculated as the difference between the Northern Hemisphere mid-latitudes and the equator (symbols and colors are as in panels (a,b)). Panel (d): scatter plot of AoA (years) versus the $N_2O$ mixing ratio (ppmv), for GEOSCCM (blue circles), ULAQ-CCM (red circles) and the median of AoA observations from Andrews et al. (2001) and Engel et al. (2009) versus $N_2O$ SMR observations (black circles). The time average is between 2001-2005. Panel (e): 50 hPa latitudinal section of the $N_2O$ mixing ratio (ppbv) from the same models and observations as in panel (d). The yellow-filled area show the range of time variability of SMR measurements (i.e., $\pm 2\sigma$).



**Table 5.** Parameters of the linear fit of polar temperatures versus eddy heat fluxes (Austin et al. (2003)). The four columns show the correlation between the heat flux at 100 hPa averaged over 40°N to 80°N for January and February versus temperatures at 50 hPa averaged over 60°N to 90°N for February and March in the Northern Hemisphere, while for the Southern Hemisphere the heat fluxes at 100 hPa are averaged between 40°S and 80°S in July and August and the temperatures at 50 hPa are averaged between 60°S and 90°S in August and September. The four columns represent, for the years 1981-2002 datas, respectively the correlation coefficient (R), the parameters for the linear fit ($T_0$ and ) and the related error for $\beta$.

| Northern Hemisphere | R | $T_0$ | $\beta$ | $\sigma$ |
|---|---|---|---|---|
| ERA40 | 0.69 | 193.8 | 1.44 | 0.27 |
| GEOSCMM | 0.80 | 193.5 | 1.65 | 0.22 |
| ULAQ-CCM | 0.65 | 192.8 | 1.29 | 0.15 |
| Southern Hemisphere | R | $T_0$ | $\beta$ | $\sigma$ |
| ERA40 | 0.83 | 188.7 | 1.04 | 0.17 |
| GEOSCMM | 0.81 | 179.3 | 2.05 | 0.32 |
| ULAQ-CCM | 0.93 | 185.4 | 1.76 | 0.29 |

In Fig. 5ab the vertical mass fluxes are evaluated by looking at the $CH_4$ and $N_2O$ measurements already used in Fig. 2 and Fig. 3 but this time combined with the vertical velocities measured by MERRA, defining the flux as [$\rho$w*]. A good agreement between measurements and models is found for the 5 to 100 hPa profile, with GEOSCCM underestimating the vertical flux between 50 and 30 hPa. In Fig. 5c we show a latitudinal breakdown of the heat fluxes averaged over the 1981-2002 period for the models and the reanalysis in order to further evaluate the transport skill of the models. A greater agreement is found over the Southern Hemisphere at mid to high latitudes compared to the Northern Hemisphere, however both models fall inside 1 $\sigma$ of the ERA40 20 years variability from 50° to 90° in both hemispheres.

## 4 Perturbation of Stratospheric Species Transport

Absorption of solar near-infrared (NIR) and planetary radiation by the geoengineering aerosols produces an increase of diabatic heating rates in the tropical lower stratosphere, resulting in local warming, changes in the latitudinal distribution of zonal winds, changes of the equatorial QBO (Aquila et al. (2014)) and a strengthening of the stratospheric Brewer-Dobson circulation (BDC) (Pitari et al. (2014)). Enhanced tropical upwelling (about 5-10% increase in vertical velocities in the lower stratosphere) and extra-tropical descent tend to move $CH_4$ poor air more efficiently towards the extra-tropical UTLS, as well as for other stratospheric long lived species. The net impact on tropospheric OH and $CH_4$ lifetime depends on the net result of superimposed species perturbations in the UTLS ($CH_4$, $NO_y$, $O_3$, $SO_4$), in addition to tropospheric chemistry perturbations due to changes in water vapor content, UV radiation and heterogeneous reactions on sulfate aerosols that affect the $NO_x$ balance. The 5-10% increase of stratospheric tropical upward mass fluxes of both $CH_4$ and $N_2O$, as shown in Fig. 6ab, is predicted



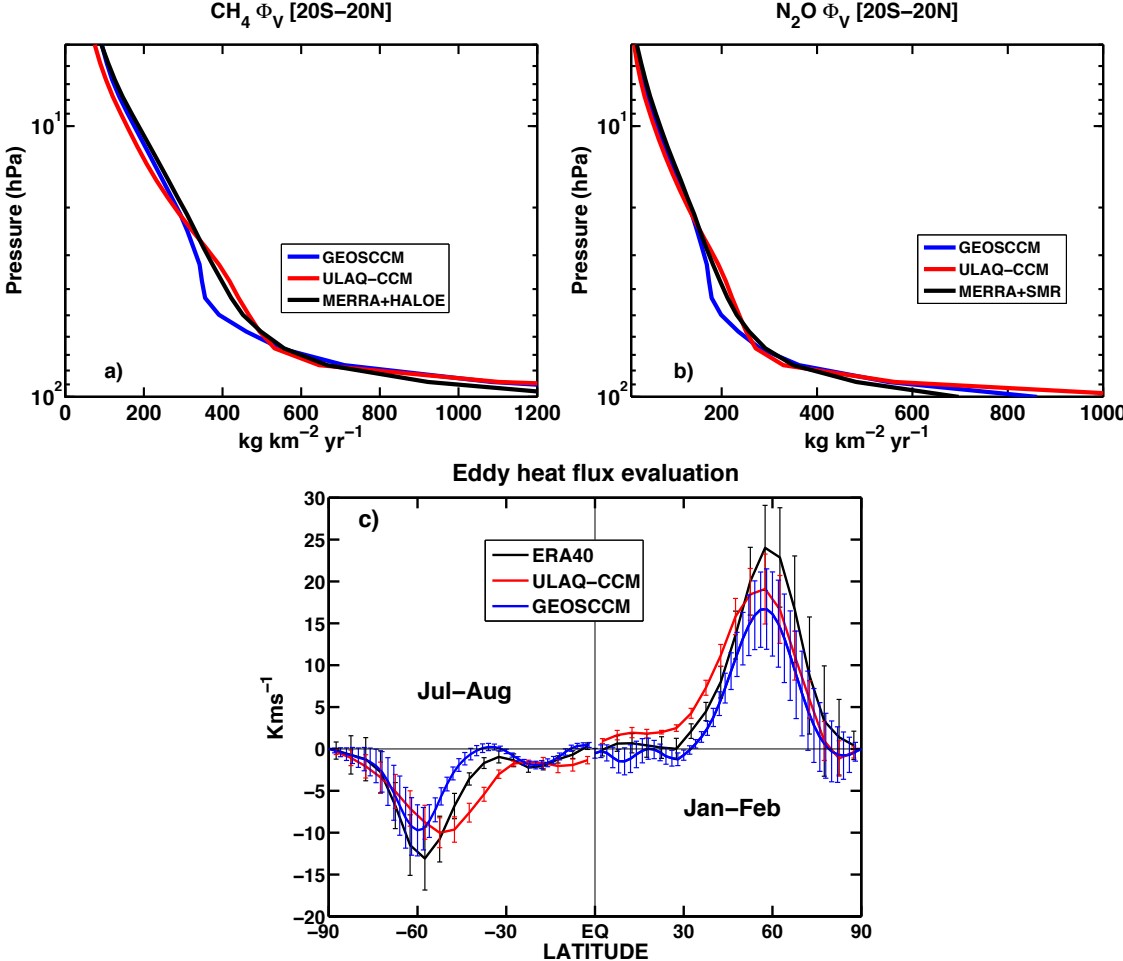

**Figure 5.** Tropical stratospheric vertical mass fluxes (20S-20N) of (a) $CH_4$ and (b) $N_2O$ for GEOSCCM (blue) and ULAQ-CCM (red) results; the vertical mass fluxes are defined as $[\rho w^*]$, where $w^*$ is the zonal mean residual vertical velocity and $\rho$ is the zonally averaged mass concentration of $CH_4$ and $N_2O$, respectively. A model evaluation is made with flux data obtained with $w^*$ from MERRA reanalysis and $CH_4$, $N_2O$ mixing ratios from HALOE and SMR results (black) (kg km$^{-2}$ yr$^{-1}$). $CH_4$ and $N_2O$ fluxes are averaged over 1991-2005 and 2001-2005, respectively, to keep consistency with the adopted HALOE and SMR mixing ratio values. Panel (c) presents an evaluation of 100 hPa horizontal eddy heat fluxes as a function of latitude averaged over 1981-2002 for the same two models (GEOSCCM in blue and ULAQ-CCM in red) with ERA40 reanalysis (Kms$^{-1}$). The eddy heat fluxes are averaged over winter months, i.e., for July and August in the Southern Hemisphere and January-February over the Northern Hemisphere and are defined as $[vT]$, where $v$ is the 3D meridional wind component and $T$ the temperature. The square brackets [] denote a zonal average and the prime a deviation from the zonal average.

by the models in geoengineering conditions, as a consequence of the increasing tropical mid-stratospheric upwelling, with a larger anomaly in GEOSCCM with respect to both MBC experiments run with the ULAQ-CCM (cases (a) and (c) in Table 1, with 8 and 5 Tg-$SO_2$ injected, respectively). The choice to only include MBC experiments when discussing vertical mass



flux anomalies is made in order to better highlight transport anomalies, because in the FBC experiment the anomaly would be largely masked by the increasing amount of tropospheric $CH_4$. The larger GEOSCCM anomaly could be explained by the QBO modification produced by geoengineering aerosols, since the prolonged lower stratospheric westerly phase produces a better tropical confinement (Trepte and Hitchman (1992); Aquila et al. (2014); Visioni et al. (2017)). This effect is absent in

the ULAQ-CCM model, which does not have an internally generated QBO, but specifies the QBO with observed equatorial zonal wind data using a nudging procedure (Morgenstern et al. (2010)).

The UTLS horizontal mixing anomalies (Fig. 6cd) are larger in case (a) of ULAQ-CCM with respect to ULAQ-CCM (c) and GEOSCCM. In the latter two model simulations, SSTs in G4 are the same as in RCP4.5, whereas ULAQ-CCM (a) is driven by SSTs taken from an atmosphere-ocean coupled model run in geoengineering conditions (i.e., CCSM-CAM4). In this case, the

larger decrease of the UTLS horizontal mixing can be explained by the increased atmospheric stabilization caused by the sea surface cooling, which is not present in GEOSCCM and ULAQ-CCM (c), where SSTs in G4 are unchanged with respect to RCP4.5. The ULAQ-CCM (c) results do not change significantly in a sensitivity simulation made increasing the stratospheric sulfur injection from 5 $Tg$-$SO_2$/yr to 8 $Tg$-$SO_2$/yr (see Table 2), pointing out to the important role of the decreasing horizontal mixing resulting from sea surface cooling, as in ULAQ-CCM (a).

The time series of model calculated $CH_4$ and $N_2O$ changes in the UTLS is presented in Fig. 7 for ULAQ-CCM and GEOSCCM. If we compare the ULAQ-CCM case (c) with GEOSCCM, the results of the two models are similar for $N_2O$ and are consistent with changes of lower stratospheric heating rates and BD circulation (due to aerosols and $O_3$). The $N_2O$ anomalies are of the order of -1 ppbv in both models (that is about -0.3%), while those of $CH_4$ are of the order of -5 ppbv

in the ULAQ model and about a factor of 2 smaller in GEOSCCM. This is due to missing chemical processes in the upper troposphere in GEOSCCM, where tropospheric OH is kept fixed at RCP4.5 values.

As already discussed in Fig. 6, the UTLS anomalies G4-RCP4.5 are rather different for ULAQ-CCM (a), mostly as a consequence of the changing SSTs in G4, with decreased horizontal mixing in the UTLS and enhanced isolation of the tropical pipe. The negative anomaly of $N_2O$ (a quasi-passive tracer) increases up to 2-4 ppbv after 2030, whereas the negative $CH_4$

anomaly increases up to approximately 10 ppbv between 2030 and 2050. A clear sign inversion is predicted after 2050 for the $CH_4$ anomaly in geoengineering conditions, as a consequence of a negative OH trend resulting from superimposed effects of $NO_x$ and $O_3$. A positive trend of stratospheric $O_3$ is, in fact, predicted in G4 with respect to RCP4.5 due to the lowering chlorine-bromine loading in the atmosphere (Pitari et al. (2014)).

The zonally averaged changes of $N_2O$ and $CH_4$ are presented in Fig. 8, with a comparison of model results from GEOSCCM and ULAQ-CCM (a). The mid-stratospheric changes are quite comparable between the two models, whereas the UTLS negative anomalies in ULAQ-CCM (a) are significantly larger for the reasons discussed above in Fig. 6-7, whereas they are fully comparable when considering GEOSCCM and ULAQ-CCM (c) results. Again, this points out to the sea surface cooling role on the UTLS horizontal mixing in sulfate geoengineering conditions.

To better understand the differences between the cases with fixed SSTs and the one with changing SSTs, in Fig. 9 we show



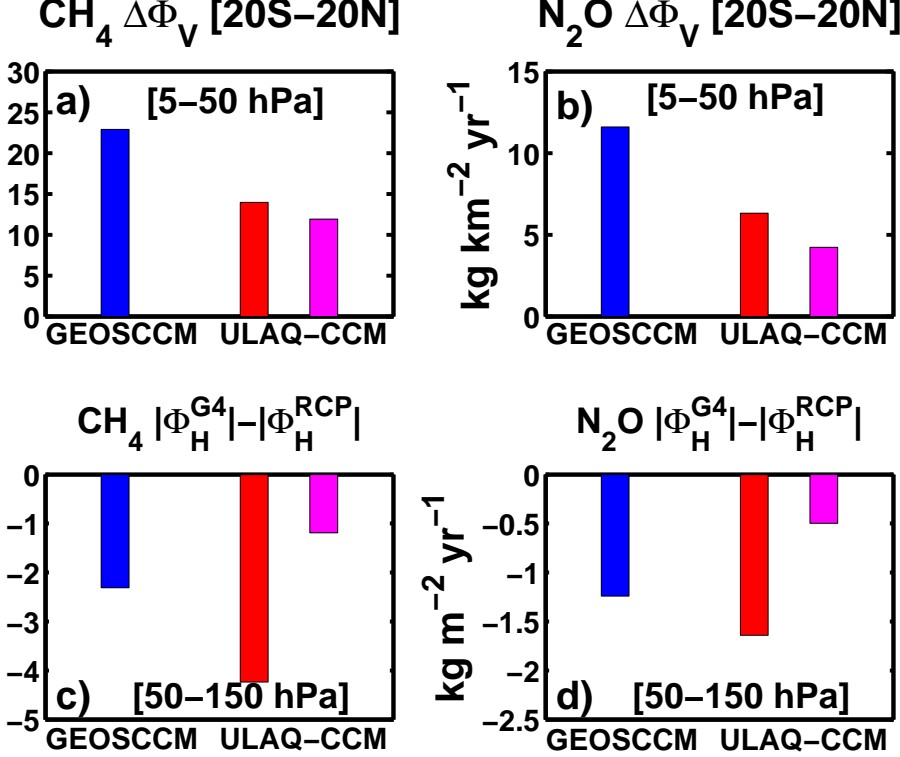

**Figure 6.** G4-RCP4.5 anomalies of (a,b) vertical and (c,d) horizontal mass fluxes of (a,c) $CH_4$ and (b,d) $N_2O$ (years 2040-49 time average). Vertical mass fluxes in panels (a,b) (defined as in Fig. 5) are averaged over the tropics (20S-20N) in the 5-50 hPa vertical layer, with GEOSCCM results in blue and ULAQ-CCM results in red and magenta for cases (a) and (c) as in Table 1, respectively (kg km$^{-2}$ yr$^{-1}$). Horizontal mass fluxes in panels (c,d) (defined as v$\rho$, with v and $\rho$ the 3D meridional wind component and mass concentration of $CH_4$ and $N_2O$, respectively) are averaged (in absolute values) over the extra-tropics (90S-20S and 20N-90N) in the 50-150 hPa vertical layer, with model results as in panels (a,b) (kg m$^{-2}$ yr$^{-1}$).

the sea surface temperatures used in ULAQ-CCM (a) and (b), which are taken from the CCSM-CAM4 ocean-atmosphere coupled model for RCP4.5 and G4 simulations (with injection of 8 Tg-$SO_2$). The time series of the globally averaged surface temperature anomalies is shown in Fig. 9a for the RCP4.5 and G4 cases: the slow oceanic response coupled to the atmospheric perturbation of long lived species delays by approximately 20 years the surface temperature return in G4 to RCP4.5 values.

5    The zonally averaged surface temperature anomalies G4-RCP4.5 are presented in Fig. 9b for the various decades from 2020 to 2090. A strong inter-hemispheric asymmetry is evident, with a negative anomaly more pronounced in the Arctic region by approximately 1.5 K with respect to the latitude range 50S-70S. The geoengineering cooling impact on Arctic sea ice is the main driver for the larger negative temperature anomaly in the Northern Hemisphere high latitudes, which favors a more pronounced atmospheric stabilization in the Northern Hemisphere winter-spring months with respect to the Southern Hemisphere.

10    The decreased horizontal fluxes of long lived species discussed in Fig. 6 for the ULAQ-CCM simulations with changing SSTs



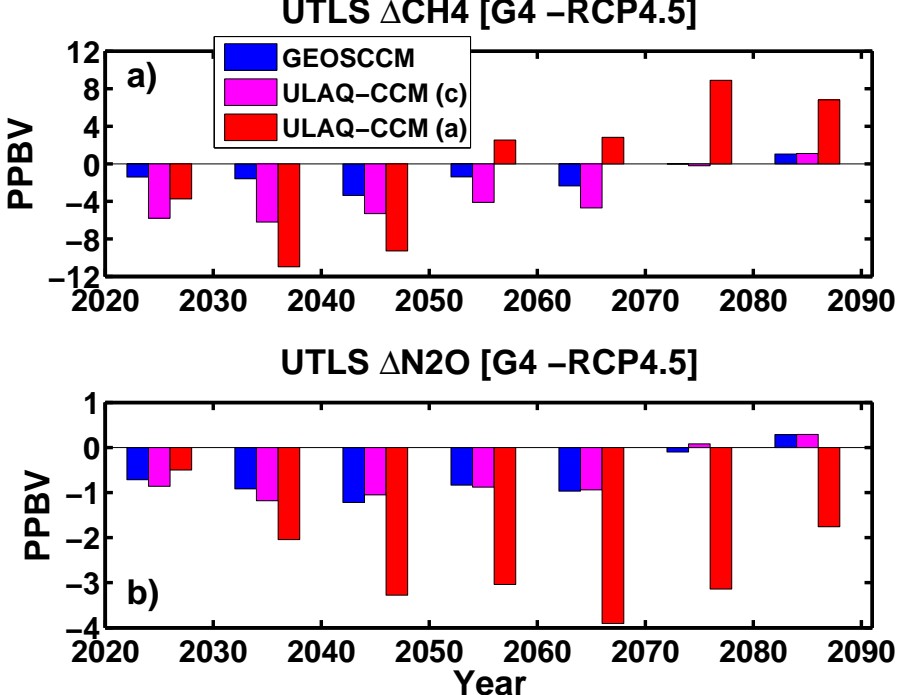

**Figure 7.** Time series of globally averaged changes of CH$_4$ (a) and N$_2$O (b) in the 50-150 hPa vertical layer, for GEOSCCM (blue) and ULAQ-CCM (red and magenta, for cases (a) and (c) as in Table 1, respectively) (decadal averages). Units are ppbv.

are a direct consequence of the atmospheric stabilization. As shown in Fig. 9c, the increased atmospheric stability in sulfate geoengineering conditions may be partially counterbalanced by the increased longitudinal variability of the induced cooling, in particular in the Northern Hemisphere, which may enhance the amplitude of planetary waves. Regions of oceanic warming in the sub-Arctic are a consequence of the increasing amount of sea ice in G4 and related enhanced transport of colder and saltier

waters towards the subpolar regions (Tilmes et al. (2009)). This favors cold sea water downwelling and thus positive anomalies of SSTs with respect to reference RCP4.5 conditions, mainly in the North Atlantic region.

Lastly, we show the anomalies of vertical and horizontal fluxes in Fig. 10 and Fig. 11, respectively, for ULAQ-CCM (a) and for GEOSCCM. For ULAQ-CCM (a), a 5% increase of the mid stratospheric tropical upward fluxes is predicted in G4 with

respect to the reference RCP4.5 case, with a pronounced inter-hemispheric asymmetry. The Southern Hemisphere increase of downward mass fluxes is much larger than in the Northern Hemisphere, both in absolute and relative units. The stratospheric mean meridional circulation is more efficiently perturbed in the Southern Hemisphere, due to the more effective atmospheric stabilization in the Northern Hemisphere (see above Fig. 9). A 5-10% decrease of the extra-tropical horizontal mass fluxes is also predicted, as expected from the discussion above for Fig. 9. The isolation of the tropical pipe is increased in a dynamical

regime with increased tropical upwelling and enhanced atmospheric stabilization. The importance of SSTs changes due to

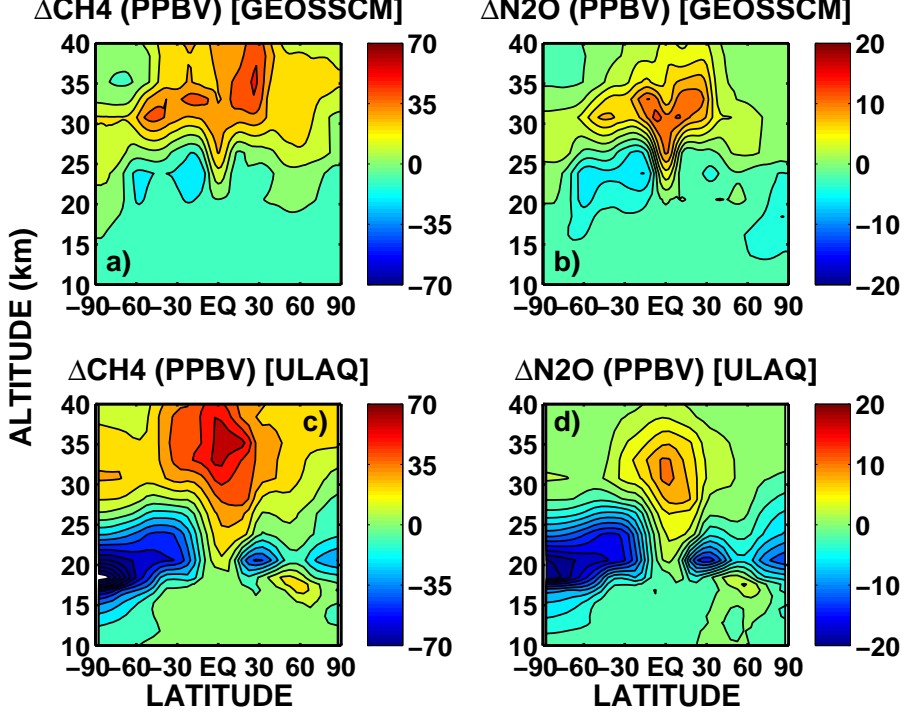

**Figure 8.** Zonal mean mixing ratio anomalies G4-RCP4.5 for (a,b) GEOSCCM and (c,d) ULAQ-CCM, for $CH_4$ (a,c) and $N_2O$ (b,d) (time average 2040-2049). ULAQ-CCM results are for case (a) in Table 1. Units are ppbv. In panels (a,c) the contour line increment is 10; in panels (b,d) the contour line increment is 2.

geoengineering is highlighted by the much smaller hemispheric difference shown by the GEOSCCM model for the downward fluxes, as well as in ULAQ-CCM (c) (not shown), while the increase in the tropical upward fluxes in Fig. 10 is comparable to the ULAQ-CCM results. Due to less atmospheric stabilization, furthermore, in Fig. 11 the GEOSCCM model shows much smaller changes in extratropical horizontal fluxes.

Another highlight of the different effects of transport and chemical effects on the lifetime is shown in Table 6, where atmospheric lifetime anomalies are shown for five species with stratospheric photolysis and O(1D) reaction, as calculated in ULAQ-CCM (b). The net lifetime changes G4-RCP4.5 result from the superposition of two effects: perturbation of species transport and sulfate aerosol induced changes in $O_3$ via $NO_x$ depletion from heterogeneous chemical reactions. The increased

10 tropical upwelling moves more efficiently these long lived species at higher altitudes in the mid stratosphere where the photolysis sink is enhanced, thus decreasing the lifetimes. On the other hand, the chemically induced ozone increase (due to the $NO_x$ sink by sulfate aerosols) tends to increase the overhead column, with a decreased mid-stratospheric UV flux. As a consequence, the photolysis rates decrease, thus prolonging the lifetimes. As shown in Pitari et al. (2014), however, the net effect on ozone



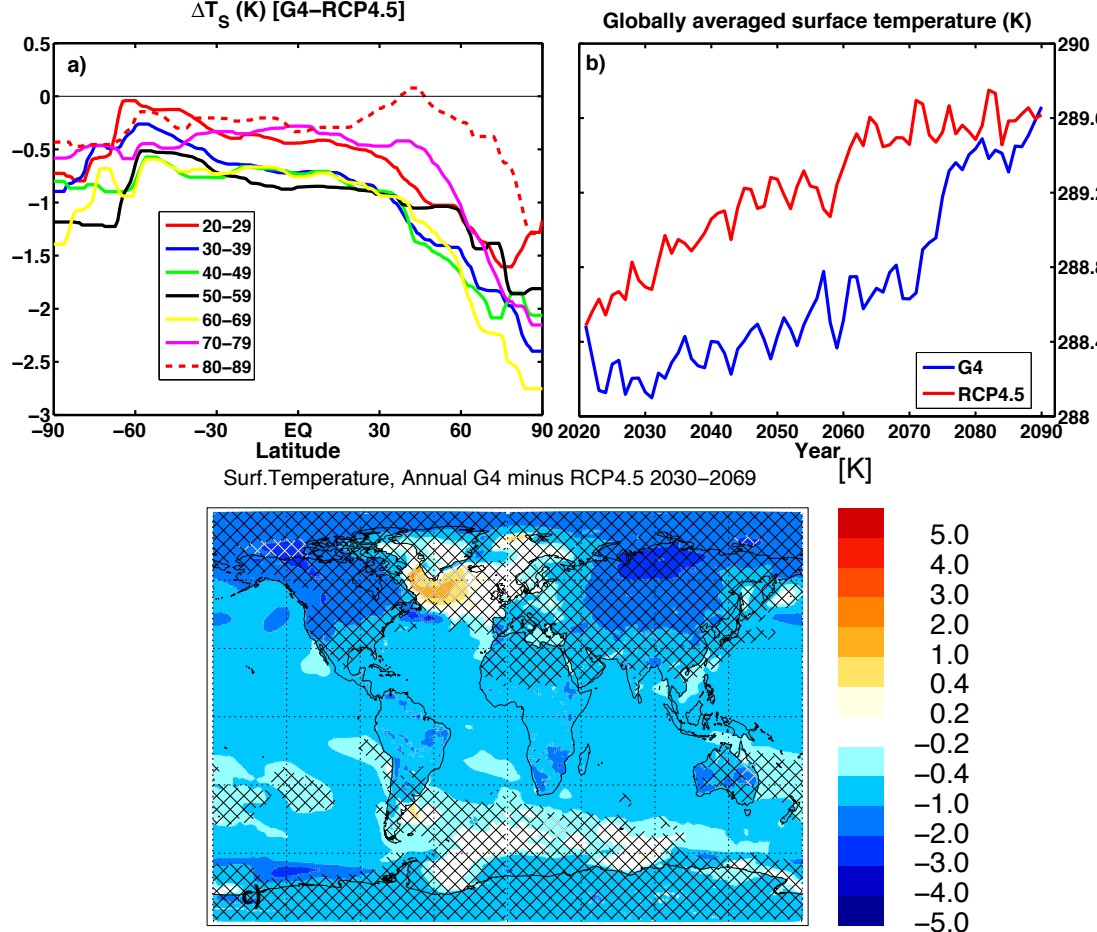

**Figure 9.** Panel (a): zonally averaged surface temperature changes G4-RCP4.5 (K) in the ULAQ-CCM (cases (a) and (b)), using sea surface temperatures from the atmosphere-ocean coupled model CCSM-CAM4 (decadal time averages from year 2020 to 2080; see legend for the different colors). Panel (b): time series of the globally averaged surface temperatures (K) from year 2020 to 2090 (RCP4.5 in red and G4 in blue). Panel (c): annually averaged surface temperature anomalies G4-RCP4.5 (K), from the atmosphere-ocean coupled model CCSM-CAM4 (time average 2030-2069). Shaded areas are not statistically significant within $\pm 1\sigma$.

of the aerosol induced $NO_x$ depletion is not constant in time, due to the decreasing amount of Cl-Br species.



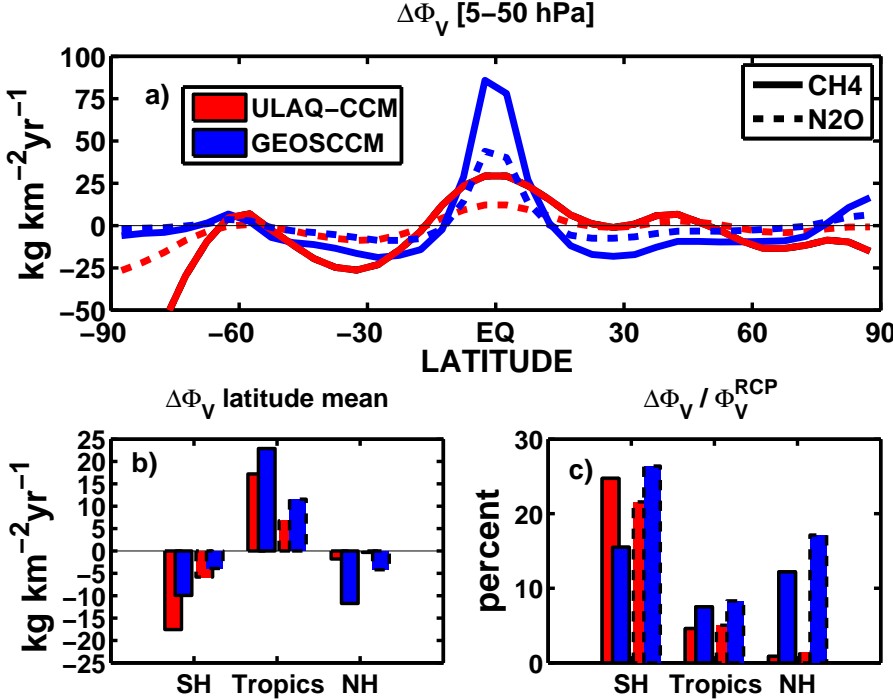

**Figure 10.** Panel (a): latitude dependent $CH_4$ (solid line) and $N_2O$ (dashed line) vertical mass flux anomalies G4-RCP4.5 from the ULAQ-CCM (a) and GEOSCCM calculations, in red and blue respectively (vertical average 5-50 hPa; time average 2040-2049). Units are kg km$^{-2}$ yr$^{-1}$. Panels (b) and (c) show the corresponding latitude averaged mass flux anomalies (absolute and percent values, respectively): SH from 90S to 20S; Tropics from 20S to 20N; NH from 20N to 90N. The vertical flux anomalies $\Delta\Phi V$ are defined as $\Delta[w*\rho CH_4]$ and $\Delta[w*\rho N_2O]$, where w* is the zonal mean residual vertical velocity, $\rho CH_4$ and $\rho N_2O$ are the mass concentrations of $CH_4$ and $N_2O$, respectively, and $\Delta$ denotes the G4-RCP4.5 difference.



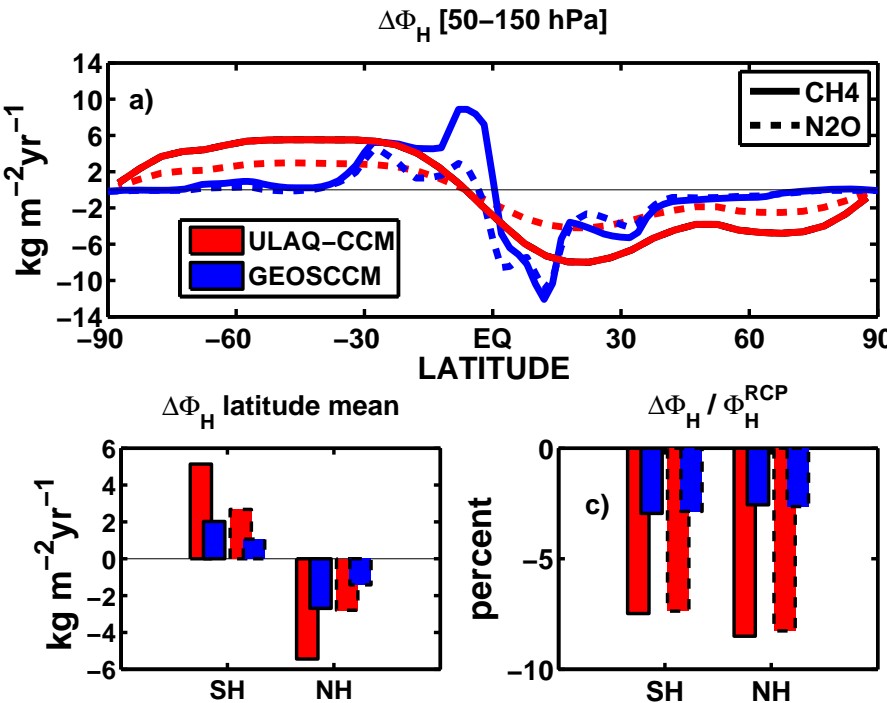

**Figure 11.** As in Fig. 10, but for horizontal mass flux anomalies G4-RCP4.5 (vertical average 50-150 hPa; time average 2040-2049). Units are kg m$^{-2}$ yr$^{-1}$. The horizontal flux anomalies $\Delta\Phi H$ are defined as $\Delta[v\rho CH_4]$ and $\Delta[v\rho N_2O]$, where v is the 3D meridional wind component.





**Table 6.** Atmospheric lifetimes (years) calculated in the ULAQ-CCM (case (b) in Table 1), relative to five species with stratospheric photolysis and $O(^1D)$ reaction sink (i.e., $N_2O$, CFC-11, H1301, CFC-12, CFC-114). First column shows year 2000 values (as an average over the 1996-2005 decade); second column shows a model mean from the SPARC (2013) report on lifetimes. Subsequent columns show the calculated lifetime anomalies due to sulfate geoengineering (average 2030-2069). Inside the square brackets we highlight the physical and chemical effects driving the lifetime changes: changing stratospheric transport in the fourth column and changing stratospheric $O_3$ in the fifth column (due to the aerosol induced $NO_x$ loss). Results in the rightmost two columns are obtained through G4 sensitivity experiments (sn1, sn3) explained in Table 2.

|  | 1996-2005 | Model Mean [SPARC, 2013] | 2030-2069 G4-RCP4.5 [All effects] | 2030-2069 G4-G4[sn3] [Transport] | 2030-2069 G4[sn1]-RCP4.5 [$NO_x \rightarrow O_3 \rightarrow UV$] |
|---|---|---|---|---|---|
| $N_2O$ | 116.1 | **115.0 $\pm$ 9.0** | -0.4 | -3.0 | +2.6 |
| CFC-11 | 52.2 | **55.3 $\pm$ 4.2** | +2.2 | -0.2 | +2.4 |
| H1301 | 77.9 | **73.4 $\pm$ 4.7** | +1.1 | -1.4 | +2.6 |
| CFC-12 | 92.0 | **94.7 $\pm$ 7.3** | -0.1 | -2.7 | +2.6 |
| CFC-114 | 202 | **189 $\pm$ 18** | -2.3 | -4.9 | +2.6 |

## 5 Perturbation of Tropospheric Chemistry

Stratosphere-troposphere exchange of geoengineering sulfate enhances the aerosol SAD in the upper troposphere, thus favoring $NO_x$ depletion through heterogeneous chemical reactions (i.e., hydrolysis of $N_2O_5$ and BrONO2) (Tilmes et al. (2009)). Again, this implies less OH production and longer $CH_4$ lifetime (mostly via $NO + HO_2 \rightarrow NO_2 + OH$). Fig. 12 compares the G4-RCP4.5 anomalies of sulfate aerosol mass and surface area density in the UTLS, as calculated in ULAQ-CCM (c) and GEOSCCM. The ULAQ model results are taken from numerical experiments (c) in Table 1 in order to make a more meaningful comparison with GEOSCCM (same injection of 5 Tg-$SO_2$/yr, SSTs in G4 with respect to RCP4.5). The ULAQ model results are taken from numerical experiments (c) in Table 1 in order to make a more meaningful comparison with GEOSCCM (same injection of 5 Tg-$SO_2$/yr, SSTs in G4 with respect to RCP4.5).

A combination of isentropic $SO_4$ transport above the tropopause and tropical upwelling/extratropical descent produces aerosol accumulation in the extratropical lower stratosphere with a clear maximum of mass density in the Northern Hemisphere (>2 $\mu gm^{-3}$ at ~12-14 km altitude). Larger values in the ULAQ-CCM of both SAD and mass density in the tropical upper troposphere are due to a more efficient gravitational settling of the particles. An important difference between the two models is that ULAQ-CCM includes an aerosol microphysical code for predicting the particle size distribution, which, on the other hand, is assigned in GEOSCCM (Pitari et al. (2014)). A similar increase of SAD is predicted by both models in the extratropical upper troposphere, ranging between 2÷10 $\mu m^2 cm^{-3}$.

The upper tropospheric increase of sulfate aerosol surface area density is the major diver for tropospheric $NO_x$ changes in





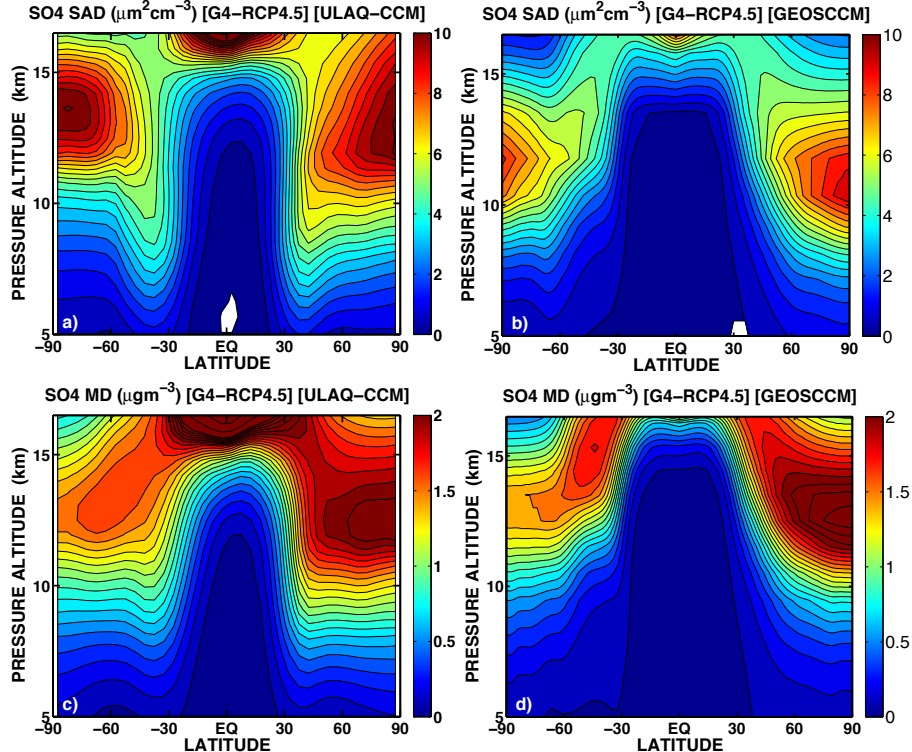

**Figure 12.** G4-RCP4.5 anomalies of sulfate aerosol surface area density (a,b) and mass density (c,d) in the upper troposphere and lowermost stratosphere, from ULAQ-CCM (a,c) and GEOSCCM (b,d) (time average 2040-2049). ULAQ-CCM results are from numerical experiments (c) in Table 1. Units are $\mu m^2 cm^{-3}$ for the surface area and $\mu gm^{-3}$ for the mass density. In panels (a,b) the contour line increment is 0.5 for values less than 12 and 2.0 from 14 to larger values. In panels (c,d) the contour line increment is 0.1 for values less than 2.5 and 1.0 from 3.0 to larger values.

geoengineering conditions. Enhanced heterogeneous $NO_x$ conversion to $HNO_3$ on the aerosol surface ends up limiting the efficiency of reaction $NO + HO_2 \rightarrow NO_2 + OH$, thus reducing OH and upper tropospheric $O_3$ production, with a consequently longer $CH_4$ lifetime. Fig. 13 shows the ULAQ model calculated anomaly of UTLS $NO_x$ in experiment (b) of Table 1, with values ranging between -0.02 and -0.2 ppbv in the upper troposphere (10÷30% reduction).

The tropospheric OH balance is also affected also by the UV amount available for $O(^1D)$ production from $O_3$ photolysis ($H_2O + O(^1D) \rightarrow 2OH$), and indirectly from the upper tropospheric $O_3$ reduction due to the decreased chemical production from $NO+HO_2$ and $NO+RO_2$. Upper tropospheric ozone, however, is also affected by perturbed strat-trop fluxes and lower stratospheric ozone depletion in geoengineering conditions (Pitari et al. (2014); Xia et al. (2017)). High latitude stratospheric

10 ozone depletion produces significant UVB increase at the surface (Tilmes et al. (2012)). On the other hand, the enhanced radiation scattering in the tropical lower stratosphere overbalances the UVB increase due to tropical stratospheric ozone losses,





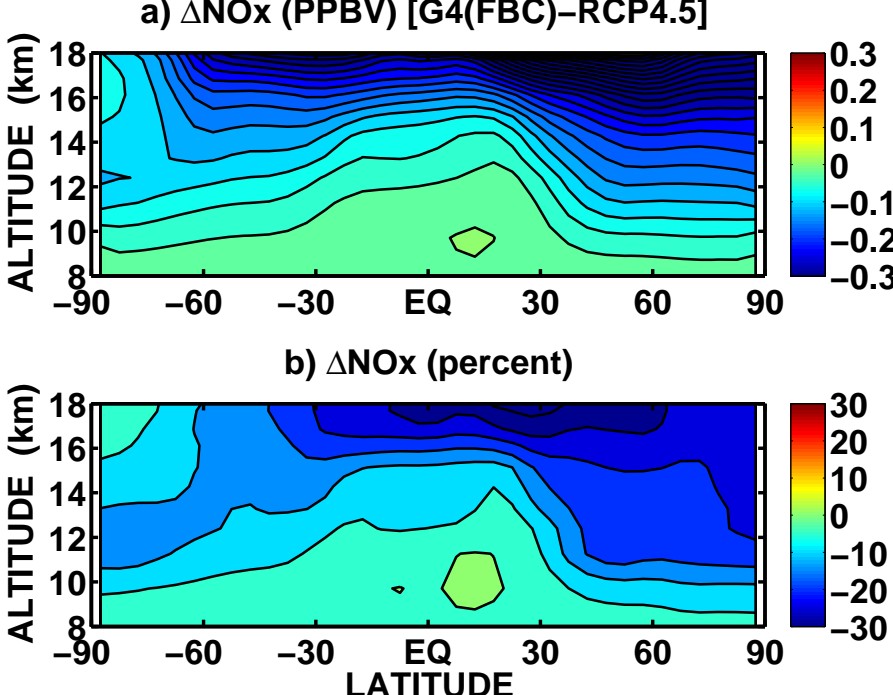

**Figure 13.** G4-RCP4.5 anomalies of NO+NO₂ mixing ratios in the upper troposphere and lowermost stratosphere, from experiment (b) of the ULAQ-CCM (time average 2040-2049). Panels (a,b) are for absolute (ppbv) and percent NO$_x$ changes, respectively. The contour line increments are 0.025 ppbv and 5%, in panels (a,b) respectively.

ending up in a net decrease of tropospheric UVB, that is again less OH production and longer CH₄ lifetime (which is mostly regulated by tropical OH). Fig. 14 shows the percent anomalies of UVB as calculated in GEOSCCM and ULAQ-CCM (c), for the two components that are explicitly on-line in the models (O₃ and sulfate aerosols). A 1.5÷2.0% UVB decrease is predicted by the models equatorward of 40 latitude in both hemispheres (-1.60% for GEOSCCM and -1.94% for ULAQ-CCM). The sul-
5   fate geoengineering impact on tropospheric UV penetration and heterogeneous chemistry changes have been widely discussed in Xia et al. (2017), along with their effects on surface ozone concentration.

Solar radiation scattering by geoengineering aerosols increases the planetary albedo and cools the surface, with a tropospheric water vapor decrease as a response to this cooling: less OH is produced by reaction H₂O + O($^1$D), thus prolonging
10  the CH₄ lifetime. The combination of this climate-chemistry effect with the others discussed above (NO$_x$, UV, strat-trop O₃ transport) produces the net OH perturbation in G4 with respect to RCP4.5 (Fig. 15a) and the resulting CH₄ change (Fig. 15b). The calculated average tropospheric anomaly of CH₄ is +190 ppbv, that is 10.6% with respect to the RCP4.5 base case average mixing ratio in years 2040-2049. The stratospheric anomalies are consistent with those discussed in Fig. 8c, obtained with the



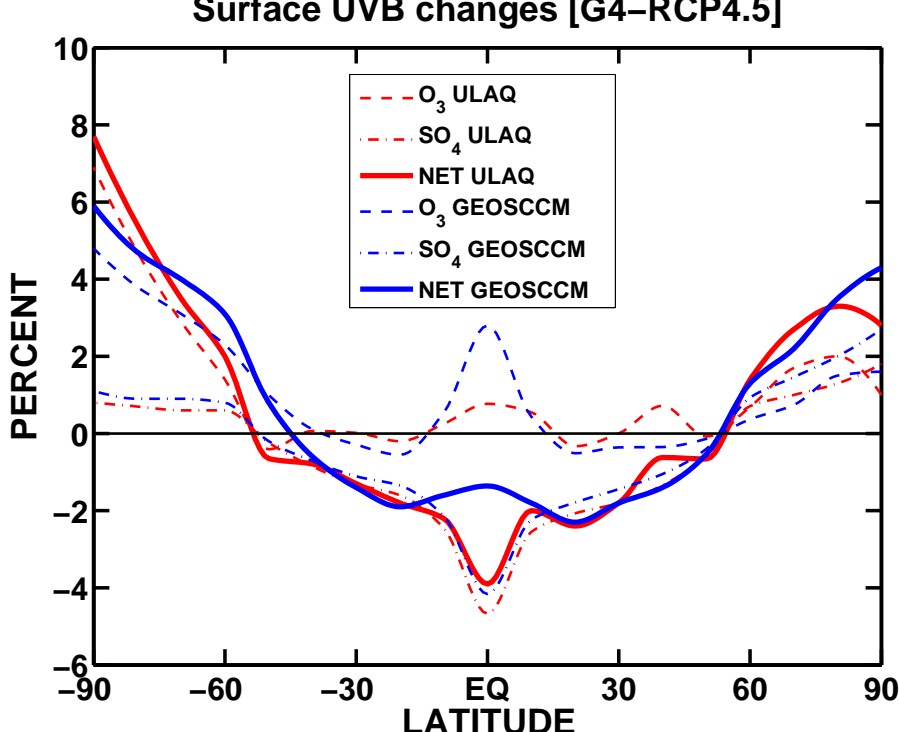

**Figure 14.** G4-RCP4.5 percent anomalies of surface UVB as a function of latitude, from ULAQ-CCM (c) (red) and GEOSCCM (blue) (2040-2049). UVB changes are shown for the two components that are explicitly on-line in the models (i.e., $O_3$ and aerosols) and for their net. ULAQ-CCM results are taken from numerical experiment (c) in Table 1 in order to make a more meaningful comparison with GEOSCCM, as in Fig. 12.

same G4 perturbation, but using the MBC approach (ULAQ-CCM (a)).

Any attempt to assess the long-term atmospheric response of $CH_4$ to OH changes needs the surface mixing ratio to be allowed to respond freely to tropospheric perturbations of its main sink process (i.e., oxidation by OH), which determines the $CH_4$ lifetime. The usual modeling approach of adopting an assigned time-dependent mixing ratio as a surface boundary condition (MBC) can still be used to calculate climate-chemistry induced changes in $CH_4$ lifetime, but this cannot provide information on the tropospheric mass changes of $CH_4$ induced by the OH perturbations. In addition, to obtain a correct estimate of the lifetime perturbation, the MBC approach would necessitate the use of correction factors, due to the missing feedback of lower tropospheric $CH_4$ changes on $HO_x$ chemistry (Myhre et al. (2011)).





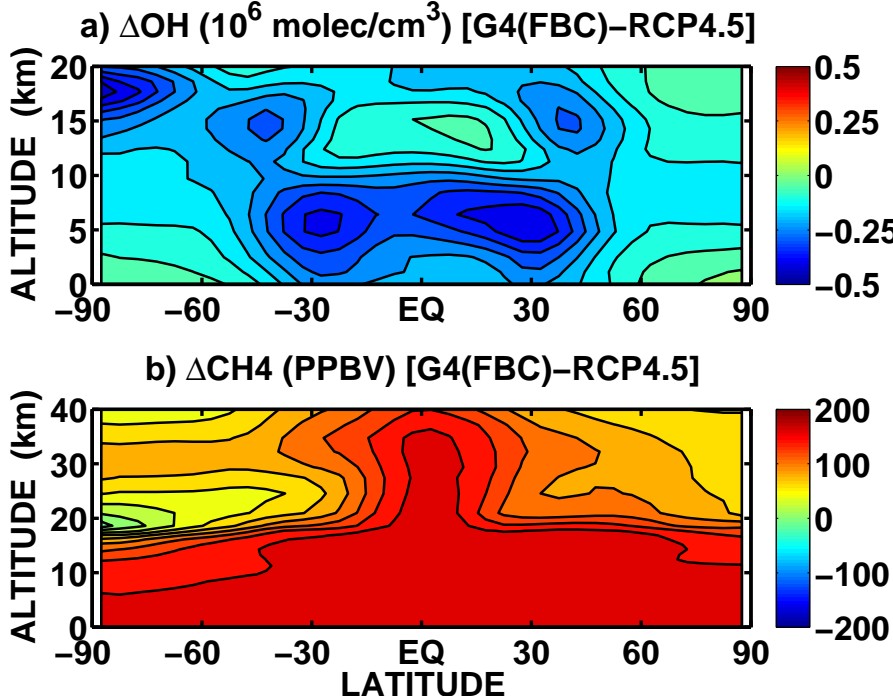

**Figure 15.** ULAQ-CCM calculated G4-RCP4.5 anomalies of (a) OH concentrations and (b) CH$_4$ mixing ratios (time average 2040-2049), from experiment (b) in Table 1. Units are 106 molec/cm$^3$ for OH and ppbv for CH$_4$. The contour line increment is 0.1106 molec/cm$^3$ for OH and 25 ppbv for CH$_4$.





**Table 7.** $CH_4$ surface emissions, sinks, global mass burden and lifetime in the ULAQ-CCM for experiment (b) (year 2000). (a) IPCC (2013); (b) Wecht et al. (2014); (c) Lamarque et al. (2010).

| Emissions (Tg/yr) Sinks (Tg/yr) Burden (Tg) Lifetime (yr) | ULAQ-CCM [FBC] |
|---|---|
| Natural sources | $230^{(a,b)}$ |
| [wetlands] | $160^{(a,b)}$ |
| [termites] | $20^{(a,b)}$ |
| [geological] | $50^{(a)}$ |
| Anthropogenic sources | $340^{(a,c)}$ |
| [agricolture] | $125^{(a,c)}$ |
| [fossil fuel] | $100^{(a,c)}$ |
| [waste] | $79^{(a,c)}$ |
| [biomass burning] | $36^{(a,c)}$ |
| Total sources | $570^{(a,c)}$ |
| Soil deposition | $30^{(a)}$ |
| Atmospheric loss [ OH O($^1$D) Cl] | 540 |
| Total skins | 570 |
| Global mass burden | 4760 |
| Atmospheric lifetime | 8.8 |
| Global lifetime | 8.35 |

The alternative approach of using surface flux boundary condition (FBC) would in principle resolve these issues. Table 7 summarizes $CH_4$ surface emissions, sinks, global mass burden and lifetime in ULAQ-CCM (b), for year 2000.The major atmospheric sink of $CH_4$ is the reaction with OH and this determines the $CH_4$ lifetime, except for an additional smaller contribution from soil deposition and an additional stratospheric sink due to $CH_4$ reactions with O($^1$D) and Cl. The calculated OH abundance is then critical in the determination of a realistic global burden and lifetime of $CH_4$. Tropospheric OH concentrations have been evaluated in Pitari et al. (2016a) using climatological values from Spivakovsky et al. (2000). In the same published work, a comparison of calculated tropospheric $CH_4$ mixing ratios is made with observations from TES/Aura radiances.

The ULAQ-CCM calculated time series of $CH_4$ lifetime and surface mixing ratio is presented in Fig. 16a, for both reference RCP4.5 and perturbed G4 cases, using the FBC approach (experiment (b) in Table 1). A simple approach was used for the time evolution of $CH_4$ emission fluxes: the geographical distribution was fixed at year 2000 values, but the net global value was linearly scaled to the ratio of RCP4.5 recommended surface mixing ratios in future years (dotted line in Fig. 16a) with



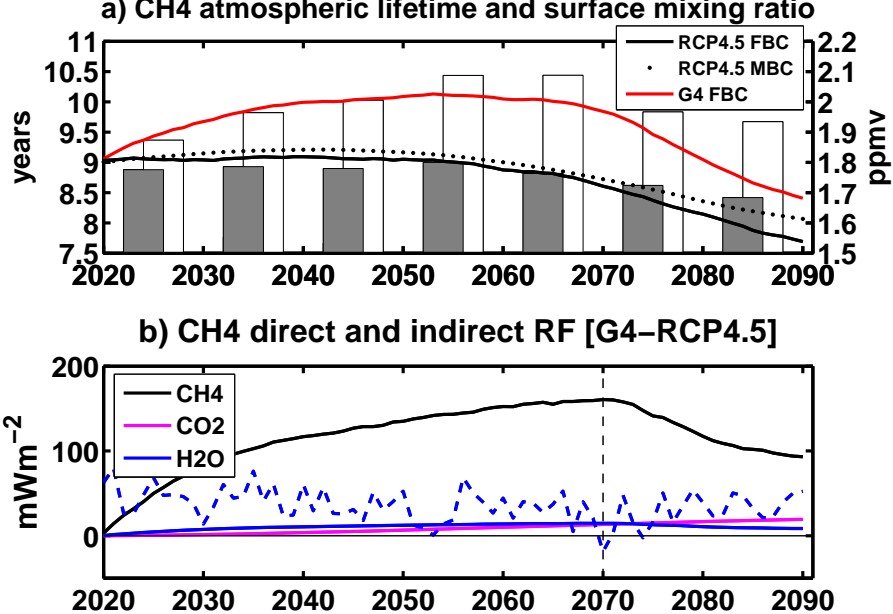

**Figure 16.** Panel (a): time series of $CH_4$ global mean lifetime (years, left scale, bars) calculated in the ULAQ-CCM FBC case (experiment (b) of Table 1), with bars referring to decadal averages (gray for RCP4.5 and white for G4). Superimposed are globally averaged $CH_4$ surface mixing ratios (ppmv, right scale), for the corresponding RCP4.5 and G4 simulations (black solid and red curves, respectively). The dotted curve shows globally averaged $CH_4$ surface mixing ratios, for the RCP4.5 MBC case (experiment (a) in Table 1), i.e., using prescribed fixed mixing ratios at the surface (Eyring and et al. (2013)). Panel (b): time series of G4-RCP4.5 radiative forcing of $CH_4$ ($mW/m^2$). Black, purple and blue curves show the direct and indirect effects (purple and blue curves are for $CO_2$ and stratospheric $H_2O$ from $CH_4$ oxidation, respectively). Dashed blue curve is for stratospheric $H_2O$ changes resulting from G4-RCP4.5 temperature anomalies at the TTL.

the year 2000 recommended value (1754 ppbv). An in-depth study of future climate change effects on $CH_4$ natural emissions or future changes on the geographical distribution of anthropogenic emissions, is beyond the purposes of the present study. The lifetime change G4-RCP4.5 shown in Fig. 16a increases up to 1.7 years in 2070 during the time period of geoengineering implementation, then slowly decreases in the so-called termination period (2070-2090), down to 1.2 years in 2090. Similarly,

5   the surface mixing ratio change increases up to 250 ppbv in 2070 and then slowly decreases in the termination period, down to 150 ppbv in 2090. These slow decreases are due to the long time needed for atmospheric $CH_4$ to return to baseline RCP4.5 values. In addition, sea surface temperatures need a few decades to recover to RCP4.5 values (Fig. 9ab), thus triggering a prolonged perturbation of the stratospheric circulation.

    A summary of gas phase RF components related to the $CH_4$ perturbation is presented in Fig. 16b. Direct stratospheric aerosol

10   RF obviously dominates in sulfate geoengineering ($\sim$ -1.2 $W/m^2$), as discussed in Visioni et al. (2017), using independent estimates available in literature. Among gas species $CH_4$ produces the largest indirect RF ($\sim$ +0.1 $W/m^2$), in addition to contributions from $O_3$ (negative) and stratospheric $H_2O$ (positive), the latter due to slight warming of the tropopause tropical



layer (TTL) (see Pitari et al. (2014)). Small indirect $CH_4$ contributions come from increasing amounts of $CO_2$ and $H_2O$ in the $CH_4$ oxidation chain. This chemical increase of $H_2O$, however, is normally smaller than the one driven by the geoengineering aerosol warming at the TTL cold point (as shown in Fig. 16b).

Table 8 summarizes our calculations for OH-dependent species lifetimes under geoengineering conditions. The ULAQ-CCM

calculated lifetimes under year 2000 conditions are fully comparable with the values in the SPARC (2013) report on lifetimes. G4-RCP4.5 anomalies averaged between 2030-2070 range between +1.33 years for $CH_4$ to +0.5 years for $CH_3CCl_3$ and +0.1 years for $CH_3Br$ and $CH_3Cl$. The FBC approach was used for $CH_4$ in order to properly evaluate its feedback on $HO_x$ chemistry. The rightmost three columns in Table 8 show the different contributions to the lifetime changes, through G4 sensitivity experiments (sn1, sn2, sn3) explained in Table 2. The major contribution to the $CH_4$ lifetime change (but also for HCFC-22

and $CH_3CCl_3$) come from the presence of aerosols with their feedback on $NO_x$-$HO_x$-$O_3$ photochemistry, as discussed before with Fig. 12-13-14 (temperature and winds are kept unchanged with respect to RCP4.5 in the G4-sn1 sensitivity case, in the chemistry module and continuity equations of chemical tracers).

The effects of tropospheric cooling with decreased water vapor (due to solar radiation scattering by the stratospheric aerosols) and strengthening of the BDC with enhanced strat-trop downward flux (due to heating rates by the stratospheric aerosols) tend

to partially or completely cancel each other. The impact of tropospheric cooling on OH-driven lifetimes is limited by the fact that the lowered $H_2O$ and OH production is partially counterbalanced by a less efficient reaction $NO+O_3 \rightarrow NO_2+O_2$ in a colder troposphere. This decreases $NO_2$ and the $NO_x$ sink to $HNO_3$, which implies an OH increase, mostly in the upper troposphere (see Fig. 17).

**Table 8.** Atmospheric lifetimes (years) calculated in the ULAQ-CCM (experiment (b) in Table 1), relative to three species that include an OH reaction sink (i.e., $CH_4$, HCFC-22, $CH_3CCl_3$). $CH_4$ is predicted with the FBC approach, the other two species with specified surface mixing ratios (unchanged between G4 and RCP4.5). First column shows year 2000 values (as an average over the 1996-2005 decade); second column shows a model mean value from the SPARC (2013) report on lifetimes. Subsequent columns show the calculated lifetime anomalies due to sulfate geoengineering (average 2030-2069). Inside the square brackets we highlight the physical and chemical effects driving the lifetime changes (see text).

|  | 1996-2005 | Model mean [SPARC (2013)] | 2030-2069 G4-RCP4.5 [All effects] | 2030-2069 G4-G4[sn2] [Temperature] | 2030-2069 G4-G4[sn3] [Transport] | 2030-2069 G4[sn1]-RCP4.5 [UV+NO₂+O₃] |
|---|---|---|---|---|---|---|
| $CH_4$ | 8.8 | **8.7 ± 1.4** | +1.33 | +0.31 | -0.28 | +1.30 |
| HCFC-22 | 10.0 | **10.7 ± 1.6** | +0.83 | +0.42 | -0.29 | +0.70 |
| $CH_3CCl_3$ | 4.6 | **4.6 ± 0.6** | +0.50 | +0.10 | -0.10 | +0.50 |

The strengthening of the Brewer-Dobson circulation affects essentially the upper tropospheric amount of $SO_4$, $CH_4$, $NO_y$ and $O_3$. This results in a negative anomaly for geoengineering $SO_4$ and for $CH_4$ (due to the enhanced lower stratospheric tropical confinement: see Fig. 11 and Fig. 8c) and a positive anomaly for $NO_y$ and $O_3$ (due to the enhanced strat-trop downward





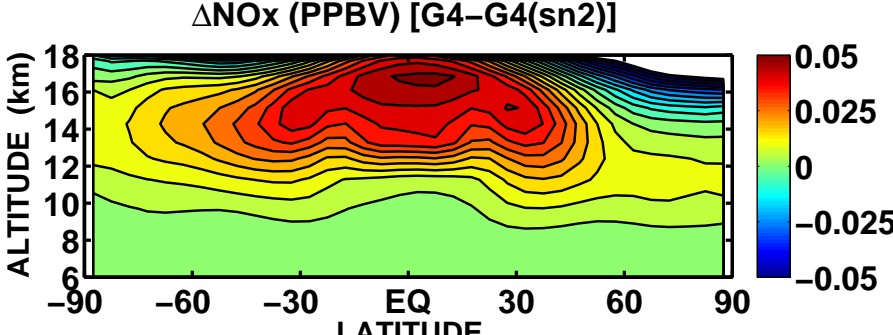

**Figure 17.** G4-G4(sn2) anomalies of NO+NO$_2$ mixing ratios in the upper troposphere and lowermost stratosphere, from ULAQ-CCM (b) (time average 2040-2049) (ppbv). The contour line increment is 0.005 ppbv. The sensitivity case G4-sn2 keeps temperature fixed at RCP4.5 values in the chemistry module.

flux). The induced OH anomaly is negative from CH$_4$ (which is an OH source) and O$_3$ (which is an OH sink in the upper troposphere). On the other hand, it is positive from SO$_4$ and NO$_y$ (due to the increasing NO$_x$ amount their negative or positive anomaly will produce). This positive NO$_x$ anomaly induced in the upper troposphere by the enhanced stratospheric circulation mostly regulates the net positive OH change in the ULAQ-CCM, with decreasing lifetimes (column five in Table 8).

## 6 Conclusions

In the present work, we have described the effect an injection of 5-8 Tg of SO$_2$ per year would have on large scale transport and lifetime of CH$_4$, using two climate-chemistry coupled models, ULAQ-CCM and GEOSCCM, both using prescribed SST coming from two atmospheric-ocean coupled models, CCSM-CAM4 and CESM, respectively. The model evaluation has shown that both models correctly simulate the vertical profiles for the chemical species under analysis (N$_2$O as a passive tracer and CH$_4$), the mean age of air and the vertical velocity w*. Furthermore, the latitudinal heat fluxes have been validated against ERA40 reanalysis in order to validate the skill of the models in correctly simulating the meridional transport.

We have shown that changes in the BDC due to lower stratospheric warming reduce the amount of CH$_4$ in the extra-tropical UTLS, both because of the strengthening of the downward branches of the BDC which brings more stratospheric air (poorer in CH$_4$) down in the upper troposphere and because of a greater isolation of the tropical pipe that reduces the amount of horizontal mixing. However, in order to properly assess the magnitude of the variations of the transport (whether it's horizontal mixing of vertical fluxes), the addition of the feedback of the ocean has proven crucial. Cooler oceans allow for a further atmospheric stabilization of the atmosphere, and the cooling of the sub-Arctic regions produces important hemispheric asymmetries that are not found in fixed SSTs simulations.

Furthermore, we have shown the changes in CH$_4$ lifetime and concentration take place because of a reduction of atmospheric OH, mostly due to three overlapping factors: reduction in tropospheric water vapor caused by the surface cooling, decrease





in O($^1$D) caused by a decrease in tropospheric UV (since part of the incoming solar radiation is scattered by the stratospheric aerosols, which also deplete ozone) and a decrease in NO$_x$ production caused by the enhancing of heterogeneous chemistry (Fig. 18). Changes in stratospheric large-scale transport and strat-trop exchange may also contribute to perturb the tropospheric amount of OH, with a net effect whose sign results from simultaneous changes of CH$_4$, NO$_y$, O$_3$, SO$_4$. All of these effects

5  may cause an increase of over 1 year for CH$_4$ lifetime in the central decades of the experiment, leading, in turn, to an increase in methane mixing ratio of over 200 ppbv.

Overall, these changes produce in our radiative transfer model calculations a positive radiative forcing of more then +0.1 W/m$^2$. While this result goes in the opposite direction with respect to the desired effect of sulfate geoengineering, it's still one order of magnitude smaller than the negative radiative forcing of the aerosols, which has been estimated to be -1.4 0.5 W/m$^2$ for a 5

10  Tg SO$_2$/yr injection, considering simulations from a vast array of models (Visioni et al. (2017)).

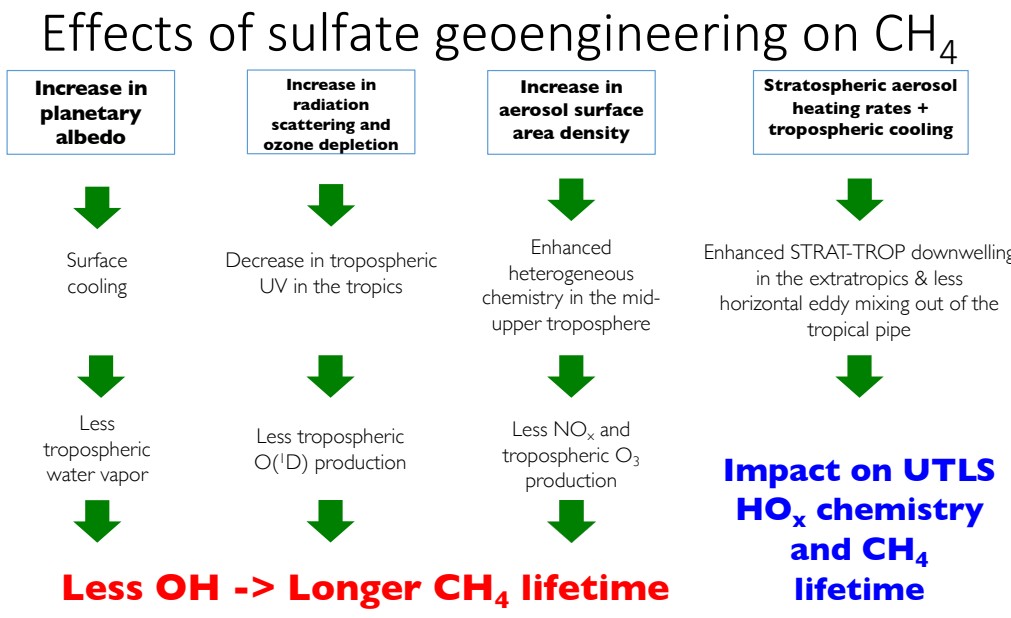

**Figure 18.** Visual representation of all contributing factors to the changes that might occur on CH$_4$ in a SG scenario.



*Data availability.* Data from model simulations are available from the corresponding author.

*Competing interests.* The authors declare no conflict of interest

*Acknowledgements.* Figure 1,2,3 and 4 were produced with the ESMValTool (Eyring et al. (2016)). GEOSCCM simulations performed by V. A. were supported by the NASA High-End Computing (HEC) Program through the NASA Center for Climate Simulations (NCCS) at Goddard Space Flight Center. I.C. acknowledges funding received from the European Union's Horizon 2020 research and innovation programme under grant agreement no. 641816 (CRESCENDO).



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
