# Peer review of "Sulfate Geoengineering Impact on Methane Transport and Lifetime: Results from the Geoengineering Model Intercomparison Project (GeoMIP)"

_Atmospheric Chemistry and Physics, 2017_

## Short Comment (SC1) · 23 Jul 2017

The manuscript acp-2017-593 proposed by D. Visioni et al, is very interesting and deserves publication.

Nonetheless the reader might feel that some important starting hypothesis to their study is missing and should be clearly indicated.

As a matter of fact, as it is written, the manuscript lets us make the assumption that the authors only considered the effects on the newly injected sulphates in the stratosphere

by the SRM technology, without taking into consideration the current tropospheric anthropogenic emissions of SO2 and their future evolution during the period in consideration.

First, we think that, with the assumption that current anthropogenic sulphur tropospheric emissions stay stable during all the period of this study, adding extra-sulphate emissions in the stratosphere would probably increase its global deposition more evenly distributed worldwide than current tropospheric emissions. Under sulphate SRM some wetlands that previously receive low amounts or did not receive tropospheric sulphates will receive (more) sulphates, and it is known that sulphate in acid rain suppresses methane emissions from natural freshwater wetlands (Gauci et al, 2008, J. Geophys. Res.), rice paddies, peat lands and other terrestrial landscapes (Oeste and al, 2107, ESD), which are the biggest methane emitters as the authors noted in table 7 of their manuscript; thus CH4 emissions reduction will occur.

Also, it is known that under a global warming (without sulphur SRM), warmer temperatures and increased rainfall in some regions will increase CH4 emissions. Under the cooling SRM scenarios envisioned by the authors (first column of figure 18 of page 30), the reverse should occur.

Two new columns in figure 18 can be added as follows:

Increase in planetary albedo => surface cooling => lower temperatures => lower CH4 emissions => lower CH4 atmospheric concentration => shorter CH4 lifetime

Increase in planetary albedo => surface cooling => lower rain fall => smaller wetlands area => lower CH4 emissions => lower CH4 atmospheric concentration => shorter CH4 lifetime

We believe the above mentioned assumption (current anthropogenic sulphur tropospheric emissions stay stable during all the studied period) should be stated in this manuscript, as:

a) current tropospheric sulphur anthropogenic emissions are and order of magnitude larger than the ones envisioned by the authors for stratospheric SRM;

b) since China's SO2 emissions started decreasing, the current trend is to a global decrease of tropospheric sulphur anthropogenic emissions (Klimont et al, 2013, Environ. Res. Lett.);

c) estimates of the amounts of sulphur pollution needed to reduce CH4 emissions of the total wetland source have been made (Gauci et al, 2004, PNAS). .

Second, the "clathrate gun hypothesis" has been debated by the scientific community as under a warming world, increased emissions from permafrost and/or from methane hydrates destabilisation is a risk. Recent work (Kohnert et al , 2017, Sci. Rep.) suggests that a new pathway of CH4 emissions exist and that it may increase if ongoing permafrost thaw continues. Under the cooling SRM scenarios envisioned by the authors the reverse should occur.

One new column in figure 18 page 30 can be added as follows: Increase in planetary albedo => surface cooling => lower temperatures => lower CH4 emissions by permafrost => lower CH4 atmospheric concentration => shorter CH4 lifetime. .

Third, we agree that the OH radical sink for CH4 is the most important in the troposphere, but it is known than the chlorine radical sink for CH4 is not only important in the stratosphere, but also occurs in the troposphere (Oeste and al, 2107, ESD), where it represents 3-5% of the CH4 removal. Variations in the tropospheric acidity may change the importance of the chlorine sink for methane. With the assumption that current anthropogenic sulphur tropospheric emissions stay stable during all the period of the author's study, adding extra-sulphate emissions in the stratosphere would probably increase the tropospheric Cl content, and, as the kinetics of the reaction of Cl radical with alkanes (including methane) are an order of magnitude larger than with the OH radical, thus the chlorine radical sink for CH4 will increase.

One new column and a new line in figure 18 page 30 can be added as follows: Increase in sulphur emissions => increased tropospheric acidity => more HCl increased Cl radical sink for CH4 => more Cl => lower CH4 lifetime

We believe that the authors should add in their manuscript that they made the assumption that this second CH4 sink (the Cl radiacal) is assumed to stay constant in their model.
* * *

---

## Author Comment (AC1) · 24 Jul 2017

The authors would like to thank R. de Richter for his general encouraging comment regarding the manuscript and for his insightful comments regarding other possible side effects of SRM on methane concentration and lifetime.

In the revised manuscript we will make sure to include in the discussion and overall conclusions the suggestions mentioned by Dr. de Richter (and update Fig. 18 accordingly) and to clarify under which assumptions our experiments have been run.

---

## Referee Comment (RC1) · Anonymous Referee #1 · 9 Aug 2017

My main issue is probably between minor and major. There is something that I think needs to be done but I hope can be accomplished without a great deal of difficulty (so sorry if the score looks severe).

My main concern with the paper is that they are discussing the impact of geoengineering using sulfate aerosol but never really show how their aerosol manifests itself. This is really crucial since if the aerosol is poorly depicted the rest of the results are essentially uninteresting. Is aerosol properly trapped at low latitudes above 20 km or does it run rapidly off to high latitudes (like it does in WACCM)? Looking at the aerosol

[Figure]

SAD anomalies, I see effectively no change in aerosol loading in low latitudes. This is at odds with what was observed after Pinatubo where a normally low aerosol region in the tropical upper troposphere is filled with aerosol for several years after the eruption (mostly due to sedimentation I suspect). In any case, I think it is critical to demonstrate that their model can produce realistic aerosol distributions for this scenario. My concern is that since they apparently see no enhancement in the tropical upper stratosphere that something unrealistic is happening with the aerosol. Please make my concerns go away.

Minor point, they seem to like to reference their own work an awful lot. This is ok but it left me with the impression that they are the only people doing key parts of this area of research.

Minor point, are they distributing the sulfur injection uniformly between 18 and 25 km? These seems impractical at best and more realistic injection scenarios would yield more realistic outcomes for aerosol distributions. Most scenarios I've seen suggest injection between 18 and 20 and counting on upward transport into the tropical pipe to distribute aerosol to higher altitudes (as observed following small and moderate eruptions and the well know water tape recorder).

Minor point, the uncertainties attached to SAGE II estimates of effective radius shown in the label for Table 1 are simply impossible or imply an impossible level of certainty in them. There are well known issues in estimating SAD with SAGE II observations at low aerosol levels which contributes to significant uncertainty in a parameter derived using it (reff). At high loading, all size discrimination of optical measurements effectively go away other than 'they are big' since the spectral dependence becomes flat and invariant for large ranges of potential sizes. Certainly the authors do not shown how they were inferred and I am wondering what they mean.

---

## Author Comment (AC2) · 10 Aug 2017

**Response to Reviewer # 1**

Reviewer's comments are in blue. Author responses are in black.

We thank the reviewer for his helpful comments, that will allow us to clarify some of the points of the manuscript.

My main issue is probably between minor and major. There is something that I think needs to be done but I hope can be accomplished without a great deal of difficulty (so sorry if the score looks severe).
My main concern with the paper is that they are discussing the impact of geoengineering using sulfate aerosol but never really show how their aerosol manifests itself. This is really crucial since if the aerosol is poorly depicted the rest of the results are essentially uninteresting. Is aerosol properly trapped at low latitudes above 20 km or does it run rapidly off to high latitudes (like it does in WACCM)? Looking at the aerosol SAD anomalies, I see effectively no change in aerosol loading in low latitudes. This is at odds with what was observed after Pinatubo where a normally low aerosol region in the tropical upper troposphere is filled with aerosol for several years after the eruption (mostly due to sedimentation I suspect). In any case, I think it is critical to demonstrate that their model can produce realistic aerosol distributions for this scenario. My concern is that since they apparently see no enhancement in the tropical upper stratosphere that something unrealistic is happening with the aerosol. Please make my concerns go away.

An in depth validation of both models regarding aerosol SAD changes due to SG and sulfate transport was already given in the Pitari et al. (2014) paper; we felt that adding a similar model evaluation would have lengthened the paper too much. However, in the reviewed manuscript we will add to Fig. 12 two additional panels (now (a-b), attached below) highlighting the aerosol SAD in the lower stratosphere for the two models, whereas the original Fig. 12 (a-b) (becoming (c-d) in the revised version) will remain to highlight the changes in aerosol SAD in the upper troposphere that are closely related to the discussion in Section 5 (tropospheric chemistry changes).

[Figure]

As can be seen by the two new panels, the aerosol distribution in both models is in agreement with several other models that have performed sulfate geoengineering simulation, and with observations after the Pinatubo eruption (for instance, SAGE II). Both models show a pronounced

confinement in the tropical lower stratosphere, with an aerosol sedimentation-driven increase of both SAD and mass density in the tropical upper troposphere, gradually approaching low values when penetrating downwards (due to irreversible removal mechanisms, namely ice particles sedimentation and wet deposition; see also Visioni et al. (2017)). A significant mid-latitude aerosol concentration is also predicted in both models due to strat-trop exchange associated to the lower branch of the Brewer-Dobson circulation. Differences between the two models in the distribution of the aerosols are due to intrinsic model differences in the size distribution (imposed for GEOSCCM and calculated for ULAQ-CCM) and the adopted radiation scheme (with impact on heating rates and hence on circulation changes). Large scale transport differences may also contribute, and the reasons are well summarized in Table 1 (treatment of QBO, SSTs, horizontal/vertical resolution). Nevertheless, both models still remain well within the range of the SAGE II measurements after the Pinatubo eruption (see Pitari et al., 2014). For good measure, we attach below a copy of Fig. 6 from Pitari et al. (2014), showing this comparison.

[Figure]

Minor point, they seem to like to reference their own work an awful lot. This is ok but it left me with the impression that they are the only people doing key parts of this area of research.

We apologize if this is the impression we have given. We have tried to include all possible published works related to the topic, and if we have failed to do so we will be glad to accept any suggestion regarding an enrichment of our bibliography. Often we cite Visioni et al. (2017) because it is a review paper where we discussed various side effects of the sulfate injection, such as effects on ozone depletion and UV changes at the surface. However, all the relevant papers presented in that one paper are also cited here when needed, and we feel none have been left out.

Minor point, are they distributing the sulfur injection uniformly between 18 and 25 km? These seems impractical at best and more realistic injection scenarios would yield more realistic outcomes for aerosol distributions. Most scenarios I've seen suggest injection between 18 and 20

and counting on upward transport into the tropical pipe to distribute aerosol to higher altitudes (as observed following small and moderate eruptions and the well know water tape recorder).

We agree that there might be more realistic injection scenarios, but the injection scenario we used is the one prescribed by the GeoMIP G4 experiment. However, we will further expand the text in the revised manuscript about the differences in injection between the two models: the GEOSCCM model injects aerosol in the 16-25 km layer in a uniform way, the ULAQ-CCM model inject the aerosol in the 18-25 km layer, but with a Gaussian distribution that puts 80% of the sulfur mass in the altitude layer from 19.5 to 22 km. This is because the GeoMIP G4 experiment suggested to inject the aerosol in a way to mimic the way any single model handles the Pinatubo eruption (Kravitz et al., 2011).

Minor point, the uncertainties attached to SAGE II estimates of effective radius shown in the label for Table 1 are simply impossible or imply an impossible level of certainty in them. There are well known issues in estimating SAD with SAGE II observations at low aerosol levels which contributes to significant uncertainty in a parameter derived using it (reff). At high loading, all size discrimination of optical measurements effectively go away other than 'they are big' since the spectral dependence becomes flat and invariant for large ranges of potential sizes. Certainly the authors do not shown how they were inferred and I am wondering what they mean.

We agree with the reviewer that there are large uncertainties in the SAD estimates with SAGE II, and we will add a caveat in the caption of Table 1. However, the values that we use for the effective radius (not the SAD, anyway) are the ones that have been made available by the SAGE group at the Langley Research Centre. We will however change the Table 1 caption in order to clarify that we are showing the standard deviation for the measurement given by SAGE II, and not an uncertainty estimated by ourselves (see also Pitari et al., 2014).

---

## Referee Comment (RC2) · Anonymous Referee #2 · 13 Aug 2017

General Comments:

This study used ULAQ-CCM and GEOSCCM to study the CH4 transport and lifetime change under sulfate injection geoengineering. The ULAQ-CCM and GEOSCCM simulation used prescribed SSTs from CCSM-CAM4. There are only a few studies working on sulfate geoengineering impact on atmospheric chemistry, and this one is important to better understand how injected sulfate aerosol will change the stratospheric circulation and CH4 chemistry. It is a good fit for ACP. However, more clarifications are needed.

[Figure]

More detailed model description are needed. It is not clear whether the models have land model coupled, and whether CH4 emission is prescribed. Better explain the experiment design, such as why use two sulfate injection amounts?

There are too many figures and tables, maybe it is better to move some of them to supplemental materials, which will also make the main text more focus on its own logistic flow.

Specific comments:

Page 1:

-Line 3: sulfate aerosol reflects and scatters the incoming solar radiation. Reflection effect should be much larger than the scattering effect in terms of increasing the planetary albedo and cooling the surface.

Page 2:

-Line 3: Please change the citation format to (Kravitz et al., 2011), and change the format through the whole manuscript.

-Line 9: delete "at visible and UV wavelengths". Solar radiation is the short wave radiation.

-Line 11: change the sentence to " a reduction of the global surface air temperature from 0.5 K (Soden et al., 2002) to 0.14 K using detrended analyses (Canty et al., 2013)"

-Line 16: please reorganize this sentence "First of all, . . .infrared wavelengths"

-Line 27: what does "a heightened exchange between the stratosphere and the troposphere" mean? The altitude change of tropopause?

-Line 28: change sentence to ". . .fluxes injects more stratosphere air into the troposphere, which dilute the troposphere concentration of such gases"

-Line 29: what is "the horizontal eddy mixing of UTLS tropical mixing ratios with the

extra-tropics"? please reorganize.

Page 3:

-Line 11 - 16: move the experiment description to session 2, and add more details of the experiment design, such as what is G4? Why there are two injection amount (5/8 Tg SO2/yr)

-Line 25-30: CCSM-CAM4 is used to provide the SST of RCP4.5 and G4 for further simulations with ULAQ-CCM and GEOCCM. Need to emphasize this. Since there are two injection amounts, maybe you should use different name for them?

Page 4:

-Line 3: what is MBC and FBC? Description needed when first appear. What's the difference among the three ULAQ-CCM experiments? How MBC and FBC make a difference?

-Table 1: Tilmes et al. (2016) used CCSM-CAM4-CHEM, which includes stratosphere and troposphere chemistry, and performed REFC1 and REFC2 experiment. You mentioned that the G4 experiment has been done with CCSM-CAM4 without interactive chemistry. This reference might be wrong for your 8Tg SO2 injection case? The CCSM-CAM4 is 40 levels? In Tilmes et al. (2016), it is 26 levels (FR) or 56 levels (SD). Why use different amount of injected SO2 in GEOSCCM and ULAQ-CCMc relative to others? Table 1: in ULAQ-CCMc and GEOSCCM, RCP4.5 SSTs are used. Does the land temperature response to the sulfate injection? In that case, would there be inconsistent between the land and the ocean?

- general question for model description: in ULAQ-CCM and GEOSCCM, is the CH4 emission prescribed? Or the two models have interactive land with dynamic vegetation and agriculture, CH4 emission is interactive with the climate? Do those two models have no ocean, and that why they need the SSTs from CCSM-CAM4?

Page 6:

**ACPD**

-Line 2: could you explain why ULAQ-CCM has a large bias over the polar region, especially in MAR? -Figure 1: why TES has much higher CH4 concentration than HALOE on both 100 hPa level and the vertical profile? Please use subscript number in chemical formulas, such as CH4. Please make this change through all plots.

-Line 12: change "significative" to "significant"

Page 7:

-Figure 2: in the figure caption, please change to "(a) and (d) 60S-90S and 60N-90N, (b) and (e) 30S-60S and 30N-60N, and (c) and (f) 30S-30N". Delete "units are ppmv". The plot itself shows the unit.

Page 9:

-Table 4: instead of confidence interval, maybe standard deviation is easier to see? (e.g. $\pm0.003$)

Page 10:

-Line 2: change to "model values"

-Line 8: delete "by knowing this"

-Line 10: change to "We looked at"

-Line 11: change to "Table 5 compares the coefficient..."

Page 11:

-Figure 4: please make sure that the tile font size is the same.

Page 13:

-Line 1: would result from MBC include both the dynamic change and the tropospheric CH4 concentration change? Would it be better to put results from MBC and FBC together, and the difference will demonstrate the dynamic change?

-Line 8: it might be better to change the sentence to something like "In the latter two model simulations, RCP4.5 SSTs are used, whereas ULAQ-CCM(a) is driven by G4 SSTs."

-Line 11: delete "where SSTs in G4 are unchanged with respect to RCP4.5."

Page 14:

-Line 20: why "missing chemical processes in the upper troposphere in GEOSCCM" only affect CH4? N2O shows similar change in two models.

-Line 33: there is no zonal vertical plot showing the comparison between GEOSCCM and ULAQ-CCM9c)

Page 15:

-Figure 6: does the difference between GEOSCCM and ULAQ-CCMc in (c) and (d) also come from the difference in QBO?

-Figure 6: the difference between ULAQ-CCMa and ULAQ-CCMc in (a) and (b), is that from the gas concentration change from the troposphere or from the tropical surface temperature difference?

-Line 1: Why showing ULAQ-CCMa and b? do those two runs both use SSTs from G4 simulation, and ULAQ-CCMc uses SSTs of RCP4.5? if the purpose is to compare SSTs in G4 and RCP4.5, then it is a comparison between ULAQ-CCM(a),(b) and ULAQ-CCM(c).

-Line 4: in Figure 9b, isn't the global averaged surface temperature back to RCP4.5 level around 2080? Then it is 10 years not 20 years.

-Line 5: the warming in North Atlantic Ocean under G4 is because the cooling in that region under RCP4.5. Please look at IPCC report, and there are observations showing the cooling in that region.

Page 16:

-Line 8: why comparing ULAQ-CCMa and GEOSCCM? ULAQ-CCMa simulates 8Tg SO2/yr injection, and used SSTs of G4, GEOSCCM simulates 5Tg SO2/yr injection, and used SSTs of RCP4.5.

Page 17:

-Figure 8: please add the pressure level on y-axis as well.

-Figure 8: figure 6 shows that the vertical mass flux in GEOSSCM is much larger than in ULAQ a and c as a result of QBO, why here the stratosphere CH4 concentration is much stronger in ULAQ? Is that because the troposphere CH4 concentration is much higher in ULAQ than in GEOSSCM?

-Figure 8: why there are strong reduction of CH4 and N2O under G4 in lower stratosphere over the south pole relative to RCP4.5 using ULAQ?

-Line 6: how was the lifetime calculated?

Page 18:

-Figure 9: In (b) G4 global averaged surface temperature returns back to RCP4.5 level around 2.80, but in (a) the red dashed line (2080-2089) shows a large negative number, with a global average close to -0.5 K. How could that be?

Page 19:

-Figure 10: the red and blue bars are overlapped.

Page 20:

-Figure 11: the red and blue bars are overlapped.

Page 21:

-Line 8-9: delete the repeat sentence.

-Line 17: what does 2Ãů10 um2 cm-3 mean? Should it be 2-10?

Page 22:

-Figure 12: what does Pressure Altitude mean? Add pressure level in y-axis.

-Line 4: 10-30%

Page 23:

-Line 1: how about over the mid-high latitude regions, UVB increases as a net result, which enhances the production of OH. In Figure 15, does the green color over pole regions on the surface mean positive or negative?

-Line 3: 1.5-2.0%

-Line 4: unit of latitude.

-Line 8: not scattering increases albedo, reflection is the main reason.

Page 27:

-Line 6: The relative long life time makes the CH4 concentration needs a longer time to return back to the RCP4.5 level after termination. But why the lifetime of CH4 need a long time to back to RCP4.5 level? Could you please explain more? How the atmospheric dynamics, the UVB, OH (which are related to the CH4 lifetime) changes after the termination?

Page 29:

-Line 6: please change to "we have described that an injection of 5-8 Tg of SO2 per year would have effect on large scale. . ."

-Line 6-8: reorganize this sentence, maybe break into two sentences.

Page 30:

-Line 1 and Figure 18: "a decrease in tropospheric UV" will be misleading. Figure 14
shows the reduction in only in tropics, and there is an increasing over mid-high latitude.
-Please discuss the uncertainty of this study, and what could be improved in future studies.

---

## Short Comment (SC2) · 4 Sep 2017

I think your results are also interesting in connection to air pollution under SRM - could you discuss the possible wider implications briefly in your conclusions? For context, see for example

Xia, L., Nowack, P. J., Tilmes, S., and Robock, A.: Impacts of Stratospheric Sulfate Geoengineering on Tropospheric Ozone, Atmos. Chem. Phys. Discuss., https://doi.org/10.5194/acp-2017-434, accepted for publication.

[Figure]

https://www.atmos-chem-phys-discuss.net/acp-2017-434/

Nowack, P. J., Abraham, N. L., Braesicke, P., and Pyle, J. A.: Stratospheric ozone changes under solar geoengineering: implications for UV exposure and air quality, Atmos. Chem. Phys., 16, 4191-4203, https://doi.org/10.5194/acp-16-4191-2016, 2016.

---

## Author Comment (AC4) · 6 Sep 2017

Thank you for your comment. We think that adding some discussion on air pollution could greatly benefit our conclusions. The first paper you mentioned is already cited in regards to ozone depletion (page 22, line 9 in the discussion paper), but will also be included in the conclusions.

We have added a paragraph in the revised manuscript at the end of page 25 (of the revised manuscript). It states: "In addition, gas species concentration changes (espe-

cially ozone) would also affect air quality and surface UV concentrations, which might have implications on human health, as already noted in Xia et al. (2017) and Nowack et al. (2016). As discussed in the present study, as well as in Nowack et al. (2016), Tilmes et al. (2012) and Pitari et al. (2014), the stratospheric ozone depletion induced by geo-engineering solar radiation management techniques directly impact the tropospheric UV budget. The health impact of surface UV increases (located only at mid-high lat-itudes in the case of sulfate geoengineering) may be partly counterbalanced by the decreased tropospheric OH concentration and O3 production."

---

## Author Response (AR2)

**Response to Reviewer # 1**

Reviewer's comments are in blue. Author responses are in black.

We thank the reviewer for his helpful comments, that will allow us to clarify some of the points of the manuscript.

My main issue is probably between minor and major. There is something that I think needs to be done but I hope can be accomplished without a great deal of difficulty (so sorry if the score looks severe).
My main concern with the paper is that they are discussing the impact of geoengineering using sulfate aerosol but never really show how their aerosol manifests itself. This is really crucial since if the aerosol is poorly depicted the rest of the results are essentially uninteresting. Is aerosol properly trapped at low latitudes above 20 km or does it run rapidly off to high latitudes (like it does in WACCM)? Looking at the aerosol SAD anomalies, I see effectively no change in aerosol loading in low latitudes. This is at odds with what was observed after Pinatubo where a normally low aerosol region in the tropical upper troposphere is filled with aerosol for several years after the eruption (mostly due to sedimentation I suspect). In any case, I think it is critical to demonstrate that their model can produce realistic aerosol distributions for this scenario. My concern is that since they apparently see no enhancement in the tropical upper stratosphere that something unrealistic is happening with the aerosol. Please make my concerns go away.

An in depth validation of both models regarding aerosol SAD changes due to SG and sulfate transport was already given in the Pitari et al. (2014) paper; we felt that adding a similar model evaluation would have lengthened the paper too much. However, in the (new) supplementary material file, we have added a new figure (Fig. S7, attached below for clarity) highlighting the aerosol SAD changes produced by SG in both models up to 25 km altitude. The original Fig. 12 (now Fig. 9 in the revised version) will remain to highlight the changes in aerosol SAD in the upper troposphere, which is closely related to the discussion in Section 5 of the manuscript (tropospheric chemistry changes.

[Figure]

As it can be seen in the two panels, the aerosol distribution in both models reflects a good isolation the tropical pipe, in agreement with observations after the Pinatubo eruption (SAGE II, for instance) and other modeling studies (Tilmes et al., 2015). Both models show a pronounced

confinement in the tropical lower stratosphere, with an increase of both SAD and mass density in the tropical upper troposphere. The latter is produced by gravitational sedimentation of the aerosol particles and gradually approaches low values when penetrating downwards, due to irreversible removal mechanisms, namely ice particle sedimentation and wet deposition (see also Visioni et al. , 2017). A significant mid-high latitude increase of the aerosol concentration is also predicted in both models in the mid-upper troposphere, due to strat-trop exchange associated to the lower branch of the Brewer-Dobson circulation.

Differences between the two models in the aerosol distribution are due to intrinsic model differences in the size distribution (imposed for GEOSCCM and calculated for ULAQ-CCM) and the adopted radiation scheme (with impact on heating rates and hence on circulation changes). Large scale transport differences may also contribute, and the reasons are well summarized in Table 1 (treatment of QBO, SSTs, horizontal/vertical resolution). Nevertheless, both models still remain well within the range of the SAGE II measurements after the Pinatubo eruption (see Pitari et al., 2014). For good measure, we attach below a copy of Fig. 6 from Pitari et al. (2014), showing this comparison for RCP4.5 and G4 conditions (SAGE values are for 1992-1993).

[Figure]

[Figure]

Minor point, they seem to like to reference their own work an awful lot. This is ok but it left me with the impression that they are the only people doing key parts of this area of research.

We apologize if this is the impression we have given. We have tried to include all possible published works related to the topic, and if we have failed to do so we will be glad to accept any suggestion regarding an enrichment of our bibliography. Often we cite Visioni et al. (2017) because it is a review paper with a rich bibliography in this area of research, where we discussed various side effects of the sulfate injection, such as effects on ozone depletion and UV changes at the surface. However, all the relevant papers presented in that one paper are also cited here when needed, and we feel none have been left out.

Minor point, are they distributing the sulfur injection uniformly between 18 and 25 km? These seems impractical at best and more realistic injection scenarios would yield more realistic outcomes for aerosol distributions. Most scenarios I've seen suggest injection between 18 and 20 and counting on upward transport into the tropical pipe to distribute aerosol to higher altitudes (as observed following small and moderate eruptions and the well know water tape recorder).

We agree that there might be more realistic injection scenarios, but the injection scenario we used is the one prescribed by the GeoMIP G4 experiment. However, we will further expand the text in the revised manuscript about the differences in injection between the two models: the GEOSCCM model injects aerosol in the 16-25 km layer in a uniform way, the ULAQ-CCM model inject the aerosol in the 18-25 km layer, but with a Gaussian distribution that puts 80% of the sulfur mass in the altitude layer from 19.5 to 22 km. This is because the GeoMIP G4 experiment suggested to inject the aerosol in a way to mimic the way any single model handles the Pinatubo eruption (Kravitz et al., 2011).

Minor point, the uncertainties attached to SAGE II estimates of effective radius shown in the label for Table 1 are simply impossible or imply an impossible level of certainty in them. There are well known issues in estimating SAD with SAGE II observations at low aerosol levels which contributes

to significant uncertainty in a parameter derived using it (reff). At high loading, all size discrimination of optical measurements effectively go away other than 'they are big' since the spectral dependence becomes flat and invariant for large ranges of potential sizes. Certainly the authors do not shown how they were inferred and I am wondering what they mean.

We agree with the reviewer that there are large uncertainties in the SAD estimates from SAGE II, and we will add a caveat in the caption of Table 1. However, the values that we use for the effective radius (not the SAD, anyway) are the ones that have been made available by the SAGE group at the Langley Research Centre. We have changed the Table 1 caption in order to clarify that we are showing the standard deviation for the SAGE II retrieval, and not an uncertainty estimated by ourselves (see also Pitari et al., 2014).

**Response to Reviewer # 2**

Reviewer's comments are in blue. Author responses are in black.

This study used ULAQ-CCM and GEOSCCM to study the CH4 transport and lifetime change under sulfate injection geoengineering. The ULAQ-CCM and GEOSCCM simulation used prescribed SSTs from CCSM-CAM4. There are only a few studies working on sulfate geoengineering impact on atmospheric chemistry, and this one is important to better understand how injected sulfate aerosol will change the stratospheric circulation and CH4 chemistry. It is a good fit for ACP. However, more clarifications are needed.

We thank the reviewer for his in depth comments on the manuscript and for its overall positive evaluation. We have tried to address all of his comments.

More detailed model description are needed. It is not clear whether the models have land model coupled, and whether CH4 emission is prescribed. Better explain the experiment design, such as why use two sulfate injection amounts?

We have tried to improve this part by expanding on our explanations of the models, in particular about the land-model coupling (neither GEOSCCM nor ULAQ-CCM have it), the prescribed emissions and the reason behind the different injection amounts.

There are too many figures and tables, maybe it is better to move some of them to supplemental materials, which will also make the main text more focus on its own logistic flow.

We have done what the reviewer suggested by moving Figure 2, 3 and 4, Table 4 and 5 and Figure 17 to a Supplementary Material. More figures has been added in the supplementary material, all of them mentioned in the single points raised by the reviewer below.

Specific comments:

Page 1:
-Line 3: sulfate aerosol reflects and scatters the incoming solar radiation. Reflection effect should be much larger than the scattering effect in terms of increasing the planetary albedo and cooling the surface.

Corrected.

Page 2:
-Line 3: Please change the citation format to (Kravitz et al., 2011), and change the format through the whole manuscript.

The citation format has been generated, together with the whole manuscript, with the Latex package provided by ACP, so we cannot change it. However, for other papers we have published on ACP we have found that the format is changed automatically to the one the reviewer suggested during the typesetting part of the process.

-Line 9: delete "at visible and UV wavelengths". Solar radiation is the short wave radiation.

Done.

-Line 11: change the sentence to " a reduction of the global surface air temperature from 0.5 K (Soden et al., 2002) to 0.14 K using detrended analyses (Canty et al., 2013)"

Changed.

-Line 16: please reorganize this sentence "First of all, . . .infrared wavelengths"

We have reorganized the sentence.

-Line 27: what does "a heightened exchange between the stratosphere and the troposphere" mean? The altitude change of tropopause?

It means that the stratospheric Brewer-Dobson circulation is intensified in its advective component (i.e., the mean meridional residual circulation), so that the strat-trop exchange is larger and more $CH_4$ and $N_2O$ poorer stratospheric air is advected into the troposphere.

-Line 28: change sentence to ". . .fluxes injects more stratosphere air into the troposphere, which dilute the troposphere concentration of such gases"

We have tried to rephrase the whole period so that its meaning it's clearer. The text now states "*An increase in the downward mid and high latitude fluxes in the lower stratosphere end up advecting more stratospheric air below the tropopause, thus decreasing the tropospheric concentration of these gases.*"

-Line 29: what is "the horizontal eddy mixing of UTLS tropical mixing ratios with the extra-tropics"? please reorganize.

It is the isentropic transport in the lower stratosphere, that moves in the extra-tropics those tracers with maximum concentration in the tropical pipe. The sentence has been simplified and reorganized. The text now states: "*In addition, the horizontal eddy mixing in the UTLS is lowered as a consequence of the atmospheric stabilization resulting from the tropospheric cooling and lower stratospheric warming, thus decreasing the isentropic transport of $CH_4$ and $N_2O$ from the tropical pipe towards the mid latitudes. This favors an additional decrease of the UTLS extratropical downward fluxes of $CH_4$ and other long-lived species (Pitari et al. (2016b)).*"

Page 3:
-Line 11 - 16: move the experiment description to session 2, and add more details of the experiment design, such as what is G4? Why there are two injection amount (5/8 Tg SO2/yr)

Done. See below for the question raised on the two injection amounts.

-Line 25-30: CCSM-CAM4 is used to provide the SST of RCP4.5 and G4 for further simulations with ULAQ-CCM and GEOCCM. Need to emphasize this. Since there are two injection amounts, maybe you should use different name for them?

CCSM-CAM4 is used only by the ULAQ-CCM model (as already stated in Table 1). We have improved Table 2 so that the amount of injection is clearer for each experiment.

-Line 3: what is MBC and FBC? Description needed when first appear. What's the difference among the three ULAQ-CCM experiments? How MBC and FBC make a difference?

Corrected. However, we feel that the difference among the three ULAQ-CCM experiments is clear from both Tables 1-2. MBC and FBC approaches for $CH_4$ make an obvious difference for the prediction of the tracer mass distribution in relation to the lifetime changes driven by OH perturbations. In the MBC approach the surface mixing ratio is assigned and fixed (it follows the time behaviour prescribed in the RCP4.5 scenario), so that the lifetime changes do not translate to a $CH_4$ mass distribution perturbation (except for a minimum amount in the upper troposphere). In the FBC approach (where surface $CH_4$ may freely respond to emission fluxes and sink processes) the tracer mass distribution responds in a coherent way to the OH-driven lifetime changes.

-Table 1: Tilmes et al. (2016) used CCSM-CAM4-CHEM, which includes stratosphere and troposphere chemistry, and performed REFC1 and REFC2 experiment. You mentioned that the G4 experiment has been done with CCSM-CAM4 without interactive chemistry. This reference might be wrong for your 8 Tg SO2 injection case? The CCSM-CAM4 is 40 levels? In Tilmes et al. (2016), it is 26 levels (FR) or 56 levels (SD).

The reference for the CCSM-CAM4-CHEM model is correct, insofar as the same model was used as in Tilmes et al. (2016), but with no interactive chemistry (as stated). However, as the reviewer correctly noticed, the number of level is wrong. The table now states the correct number of levels, which is 26.

Why use different amount of injected SO2 in GEOSCCM and ULAQ-CCMc relative to others?

The experiments with 5 Tg-$SO_2$ were the same used for Pitari et al. (2014). The ULAQ-CCM experiments with 8 Tg-$SO_2$ were performed in order to be consistent with G4-RCP4.5 SST changes from CCSM-CAM4, which were calculated under this larger injection hypothesis.

Table 1: in ULAQ-CCMc and GEOSCCM, RCP4.5 SSTs are used. Does the land temperature response to the sulfate injection? In that case, would there be inconsistent between the land and the ocean?

For GEOSCCM and ULAQ-CCM (c), land temperatures respond to the sulfate injection (as well to all other changes in radiatively active gases and particles), and yes, this does mean that there is an inconsistency. However, this is a common problem with AMIP style simulations. We remind the reviewer that CCMs, by definition, are atmospheric models which make climate-radiation-dynamics-chemistry (and sometimes aerosols) interact explicitly on-line, but sea surface temperatures have to be prescribed using observed data (for the past) or model predictions (for future years), based on independent predictions from AOGCMs or ESMs. In ULAQ-CCM (a) and (b) both land and sea surface temperatures are taken from CCSM-CAM4, for both RCP4.5 and G4 experiments. We agree with the reviewer that this is an important point, and Table 1 has been changed accordingly.

- general question for model description: in ULAQ-CCM and GEOSCCM, is the CH4 emission prescribed? Or the two models have interactive land with dynamic vegetation and agriculture, CH4 emission is interactive with the climate? Do those two models have no ocean, and that why they need the SSTs from CCSM-CAM4?

ULAQ-CCM and GEOSCCM are both, as the name suggest, CCMs. Because of this, they have no interactive ocean (as explained in the previous point), so they need SSTs from atmosphere-ocean coupled models. Just as a brief note for the reviewer: the strength of CCMs is in a highly-detailed coupling of photochemistry, radiation and climate, with a level of complexity normally higher than in coupled atmosphere-ocean models. These two CCMs, in the specific version used in the present study, are used without an explicitly interactive land-atmosphere module. Our study focuses on photo-chemically induced changes in long-lived species transport and lifetime. Future changes in land properties, potentially affecting the emission fluxes (of $CH_4$, in particular), are not taken in account because they are beyond our present purposes. We have tried to talk more in the model description about some model features, in order to clarify some of the points raised. However, we think that the reviewer poses an important point regarding the limitations of this study, so we have added a paragraph in the conclusion regarding the specific purpose of this study and its limitations, together with our opinion on how more complex models could further improve the analyses we have done.

We have also added, both in the model description section and in the conclusion, more references to published papers where the skills of both models where evaluated, in particular Morgenstern et al. (2017) (general up-to-date description of the models) and other related to the CCMVal-2 inter-comparison exercise.

Page 6:
-Line 2: could you explain why ULAQ-CCM has a large bias over the polar region, especially in MAR?

It is due to a combination of insufficient advective high-latitude downwelling (mean meridional residual circulation) and too strong eddy mixing, in the Southern Hemisphere high latitudes during the autumn season following the vortex break-up in November-December. This same explanation is now given in the manuscript.

-Figure 1: why TES has much higher CH4 concentration than HALOE on both 100 hPa level and the vertical profile?

This has been explained in depth in Pitari et al. (2016a). This is what the authors had to say regarding the specific issue raised by the reviewer:

"Annually averaged zonal $CH_4$ mixing ratios from the FBC experiments are presented for the models, along with observations from the Aura TES thermal infrared radiances at λ=8 μm, corrected using co-retrieved $N_2O$ estimates (Worden et al., 2012). The tropopause signature is well captured in the FBC model predictions, with a sudden $CH_4$ decrease due to downward transport of $CH_4$-poor stratospheric air in the downwelling branch of the extra-tropical Brewer-Dobson circulation. The tropospheric inter-hemispheric asymmetry is reasonably represented in the

models, whereas the positive vertical gradient of mixing ratios in the tropics and in the Southern Hemisphere is not replicated in model predictions. However, as discussed in Worden et al. (2012), a significant bias was found in the TES-retrieved $CH_4$ values in the upper troposphere with respect to the lower troposphere. A large part of this bias was adjusted by the TES team applying a correction that is based on co-retrieved $N_2O$ estimates. After correction, a residual of 2.8% bias was still found in the upper troposphere relative to the lower troposphere.

A quantitative point-by-point spatial evaluation of the model results for the FBC case is also presented, where HALOE data (Grooss and Russel, 2005) are used for the lower stratosphere, Aura TES satellite observations for the troposphere and both datasets for the tropical upper troposphere and extra-tropical lowermost stratosphere. An average inter-hemispheric difference of 7.5% is calculated in the mid-troposphere, which is ~50% larger than the observations, which show an average of 5% inter-hemispheric difference. Tropospheric mixing ratios in the Southern Hemisphere are underestimated in the models by approximately 50 to 100 ppbv. This may be attributed to a slower horizontal eddy mixing in the tropical troposphere, with respect to real atmosphere. However, considering also the above discussed residual positive bias of the Aura/TES upper tropospheric $CH_4$, we may conclude that the inter-hemispheric gradient in the models is roughly consistent with observations, in their ±1σ variability interval. By comparing the TES data with HALOE data, the residual bias of TES-retrieved upper tropospheric $CH_4$ mixing ratio is clearly visible. The models are generally within the HALOE data's 1σ uncertainty interval and thus, showing that the models have a good ability in capturing the strong horizontal gradient in the lower stratosphere, pointing out a good isolation of the tropical pipe in the models."

We feel that repeating this discussion would go beyond the scope of the present paper and lengthen it further, however we have added the reference to Pitari et al. (2016a) to that specific point of the paper.

Please use subscript number in chemical formulas, such as CH4. Please make this change through all plots.

We feel that not using subscript numbers in the plots is graphically much better and improves readability, because it does not take up too much space. We prefer to keep the plots in this way.

-Line 12: change "significative" to "significant"

Done.

Page 7:
-Figure 2: in the figure caption, please change to "(a) and (d) 60S-90S and 60N-90N, (b) and (e) 30S-60S and 30N-60N, and (c) and (f) 30S-30N". Delete "units are ppmv". The plot itself shows the unit.

Done.

Page 9:  -Table 4: instead of confidence interval, maybe standard deviation is easier to see? (e.g. ±0.003)

The confidence interval is not plus/minus the standard deviation, as it is mentioned at page 8, line 13. We calculate the Pearson correlation coefficient r for n pairs of independent points. Since the

sampling distribution of Pearson's r is not normally distributed, the Pearson r is converted to Fisher's z-statistic and the confidence interval is computed using Fisher's z. An inverse transform is then used to return to r space. These are the step of this procedure, with rlow and rhi being our confidence interval.

Fisher Z transform:
z=0.5* log((1+r)/(1-r))

Standard error of Z statistic:
stde=1.0/sqrt(n-3)

low and hi values (95% confidence):
zlow=z-1.96*stde
zhi=z+1.96*stde
inverse z-transform return to r space
rlow=(exp(2*zlow)-1)/exp(2*exp(2*zlow)+1)
rhi=(exp(2*zh1)-1)/exp(2*exp(2*zhi)+1)

Page 10:
-Line 2: change to "model values"

Done.

-Line 8: delete "by knowing this"

Done.

-Line 10: change to "We looked at"

Done.

-Line 11: change to "Table 5 compares the coefficient. . ."

Done.

Page 11:
-Figure 4: please make sure that the title font size is the same.

Done.

Page 13:
-Line 1: would result from MBC include both the dynamic change and the tropospheric CH4 concentration change? Would it be better to put results from MBC and FBC together, and the difference will demonstrate the dynamic change?

The purpose of Figures 6 to 11 is to compare the anomalies of the two models related to changes in dynamics, to see how much they are consistent and to finally discuss the reasons of inconsistencies. To do this we can only work on MBC cases. The FBC approach is closely connected

with tropospheric chemistry issues and this is the reason why the results of ULAQ-CCM (b) are only used in the final section of the paper.

-Line 8: it might be better to change the sentence to something like "In the latter two model simulations, RCP4.5 SSTs are used, whereas ULAQ-CCM(a) is driven by G4 SSTs."

Changed.

-Line 11: delete "where SSTs in G4 are unchanged with respect to RCP4.5."

Removed.

Page 14:
-Line 20: why "missing chemical processes in the upper troposphere in GEOSCCM" only affect CH4? N2O shows similar change in two models.

Because OH only reacts with $CH_4$ in the troposphere, not $N_2O$, thus the differences in OH do not affect $N_2O$ and the changes are similar between models.

-Line 33: there is no zonal vertical plot showing the comparison between GEOSCCM and ULAQ-CCM (c)

We have decided not to show ULAQ-CCM (c) because in Figure 8 we are trying to discuss the differences between the simulations we decided to compare in Figure 10 and 11. However, in the supplementary material we have added a comparison between GEOSCCM, ULAQ-CCM (c), and ULAQ-CCM (a), as per the reviewer request. The figure is below.

[Figure]

Page 15:
-Figure 6: does the difference between GEOSCCM and ULAQ-CCMc in (c) and (d) also come from the difference in QBO?

Yes, considering the similarities in the control runs (Figure 5c), the conclusion is that these differences likely come from differences in QBO. We have added this discussion in the manuscript.

-Figure 6: the difference between ULAQ-CCMa and ULAQ-CCMc in (a) and (b), is that from the gas concentration change from the troposphere or from the tropical surface temperature difference?

First of all, the difference is small. Second, it is produced by non-linear combination of the two anomalies (tropical $CH_4$ and $N_2O$ concentrations and w*). But we do not think this point deserves in-depth discussion, because the anomalies are really comparable to each other. Contrary to the UTLS anomalies in the extra-tropics (SST effect) and to the much larger GEOSCCM tropical anomaly (QBO effect).

-Line 1: Why showing ULAQ-CCMa and b? do those two runs both use SSTs from G4 simulation, and ULAQ-CCMc uses SSTs of RCP4.5? if the purpose is to compare SSTs in G4 and RCP4.5, then it is a comparison between ULAQ-CCM(a),(b) and ULAQ-CCM(c).

Yes, we intended to say that we were comparing SSTs used in (a) and (b) versus the SSTs used in (c) (and the control case). We have changed this in the manuscript to make it clearer.

-Line 4: in Figure 9b, isn't the global averaged surface temperature back to RCP4.5 level around 2080? Then it is 10 years not 20 years.

There are still differences in that decade. Also by looking at the SSTs curve in Figure 9a, we can see that the differences are almost the same as the ones for the initial decade of the experiment. If we consider the "termination period" to be a phase of equilibrium similar to the control case, then two decades seem to be a better estimate than one. However, we have changed the text to say "more than one decade".

-Line 5: the warming in North Atlantic Ocean under G4 is because the cooling in that region under RCP4.5. Please look at IPCC report, and there are observations showing the cooling in that region.

We acknowledge that, in the IPCC report, there is a cooling of that particular region. And we do expect that, for the same reasons stated in the report, in case of sulfate geoengineering that region would warm. The reason for this is that an increase of sea-ice under geoengineering (because of the surface cooling) would produce saltier waters, which being denser would go deeper when moving south, allowing for warmer sea surface temperatures. So when these two effects combine, we obtain the warming we see in the anomalies that we are discussing. We have added a short explanation for what happens in the RCP4.5 scenario in the manuscript.
However, comparing the IPCC report to our experiment is, in a way, wrong. The IPCC report studies the time evolution of surface temperatures under a given emission scenario (for example RCP4.5) and the anomalies are time-anomalies (i.e., future years versus present time). On the other hand, in our study on the potential SG impact, the anomalies are among two different scenarios at a given time horizon, so that the fact that under RCP4.5 that region of the North Atlantic is cooling down, cannot explain why CCSM-CAM4 predicts a warming under SG. The explanation we propose (based on differences in deep water formation) may however apply to both anomalies.

Page 16:
-Line 8: why comparing ULAQ-CCMa and GEOSCCM? ULAQ-CCMa simulates 8Tg SO2/yr injection, and used SSTs of G4, GEOSCCM simulates 5Tg SO2/yr injection, and used SSTs of RCP4.5.

We wanted to highlight the changes due to the inclusion (or not) of varying SSTs, while also comparing differences between the two models. We have decided not to include the ULAQ-CCM

(c) results in Figures 10 and 11, because the inter-model differences were not as important in this case. The inclusion of varying SSTs produces much larger differences (mainly in the UTLS) with respect to those attributable to 8 or 5 Tg-$SO_2$ injection. For the sake of completeness, we have included in the Supplementary material a direct comparison of long-lived species anomalies due to changes in dynamics, for GEOSCCM, ULAQ-CCM (c) and ULAQ-CCM (a), as mentioned above. Furthermore, we have added a figure in the supplementary that is the same as Figure 11 in the manuscript, but including also the values for ULAQ-CCM (c) in it. While we still think that the comparison between GEOSCCM and ULAQ-CCM (a) is the focus of the section, we believe that adding ULAQ-CCM (c) to the supplementary material could help convince any reader that, like the reviewer, might have the same doubts. The figure is below.

[Figure]

Page 17:
-Figure 8: please add the pressure level on y-axis as well.

We would like to answer here this reviewer request, made also ahead in his comment. Pressure altitude (that we sometime refer to as simply Altitude) is the atmospheric altitude calculated with a fixed scale height (7 km in our case), so that it is not a geometric altitude, but a way to show a log-pressure scale. Anyway it is a classical definition, with plenty of examples in the literature. We prefer to stick with pressure altitude on the y axis, which is indeed pressure: only a log operation has to be done.

-Figure 8: figure 6 shows that the vertical mass flux in GEOSSCM is much larger than in ULAQ a and c as a result of QBO, why here the stratosphere CH4 concentration is much stronger in ULAQ? Is that because the troposphere CH4 concentration is much higher in ULAQ than in GEOSSCM?

Figure 6 shows the vertical mass flux anomaly averaged from 5 to 50 hPa and from 20S to 20N. It is a mass flux anomaly, so that it is obvious that in the average the vertical layer closer to 50 hPa has a much higher relative importance with respect to the upper part. In the lower layer there are clear regions of negative tracer anomalies in the ULAQ-CCM (a) case, for the reasons widely discussed in the manuscript. In addition, it is important to remind that Fig. 6 compare fluxes, whereas Fig. 8 shows the final changes produced on the tracers distribution, which are a function of the flux divergence, with coupling of vertical and horizontal motions. The flux anomalies presented in Fig. 6 are essentially (but not exactly) a proxy of the w* anomalies, that are larger in GEOSCCM mainly for the different treatment of the QBO (internally generated in this model). A figure from Pitari et al. (2014) is included below, which summarizes SG induced w* anomalies among different models (including GEOSCCM and ULAQ-CCM).

[Figure]

-Figure 8: why there are strong reduction of CH4 and N2O under G4 in lower stratosphere over the south pole relative to RCP4.5 using ULAQ?

We have explained the reason in the discussion for Figures 9, 10 and 11. In particular, we note that the stronger reduction of both lived species in the Southern Hemisphere is caused by a more efficient perturbation in the stratospheric mean meridional circulation. The greater the increase of descent (Fig. 10), the greater the penetration of long lived species poorer air in the UTLS.

-Line 6: how was the lifetime calculated?

The atmospheric lifetime is calculated the way is supposed to, i.e., integrated tracer mass in the atmosphere divided by the integrated chemical loss of the tracer. We start from daily values of

both quantities, we divide them and finally we average the daily results over the time period considered. Taking into account that this is a textbook definition, and that there are no others, we do not feel that there is the need to add any further description in the manuscript.

-Figure 9: In (b) G4 global averaged surface temperature returns back to RCP4.5 level around 2080, but in (a) the red dashed line (2080-2089) shows a large negative number, with a global average close to -0.5 K. How could that be?

We think the problem might be on how the global average is considered. If the global average on Fig. 9a is performed as a weighted (by the cosine of latitude) average (as it should be done), then the number is not so close to -0.5 K, and is indeed less than 0.3 K. This is the same value visible in Fig. 9b, when the whole decade is considered.

  -Figure 10: the red and blue bars are overlapped.
  -Figure 11: the red and blue bars are overlapped.

This is a choice we made considering both readability and the presence of a finite space. We feel that our choice is correct and we would like to stick to it.

  -Line 8-9: delete the repeat sentence.

Done, thank you.

-Line 17: what does 2Ãu̇ 10 um2 cm-3 mean? Should it be 2-10?

Yes, we meant the range 2 to 10. We clarified.

-Figure 12: what does Pressure Altitude mean? Add pressure level in y-axis.

See the reply above regarding Figure 8.

-Line 4: 10-30%

Corrected.

-Line 1: how about over the mid-high latitude regions, UVB increases as a net result, which enhances the production of OH.

It is meaningless to attempt a one-by-one correlation between tropospheric UVB and OH changes. We have spent a good part of the manuscript to highlight that there is an overlap of causes determining the final net OH change, in any given part of the atmosphere. In the troposphere the UVB balance is important, but very important are also the budget of $H_2O$ (which is a function of temperature) and the budget of NOx, which is a function of aerosol SAD and interactions with other chemical species. In addition, there are effects of $O_3$, $CH_4$ itself and so on. Taking the net

increase of mid-high latitude UVB (mainly produced by the stratospheric $O_3$ depletion) to find a corresponding increase in OH, is simply meaningless.

In Figure 15, does the green colour over pole regions on the surface mean positive or negative?

The green colour is the 0 level. Please consider also what is written in the point above.

-Line 3: 1.5-2.0%

Corrected.

-Line 4: unit of latitude.

Corrected.

-Line 8: not scattering increases albedo, reflection is the main reason.

Corrected. However, since the reviewer has pointed this out before, we would like to specify that reflection is nothing else but backward scattering, so every time we mentioned "scattering" we were not, per se, wrong, although surely reflection is more precise and we have changed it everywhere as suggested.

Page 27:
-Line 6: The relative long life time makes the CH4 concentration needs a longer time to return back to the RCP4.5 level after termination. But why the lifetime of CH4 need a long time to back to RCP4.5 level? Could you please explain more? How the atmospheric dynamics, the UVB, OH (which are related to the CH4 lifetime) changes after the termination?

The lifetime is defined as integrated tracer mass divided by the integrated chemical sink of the tracer (both in the whole atmosphere). Now if a tracer has a lifetime of approximately 10 years and, for example, we switch off all the tracer sources at the surface, the lifetime would act as an e-folding time and we would need to wait for a time much longer of 10 years to arrive to a global atmospheric mass being for example 5% of the initial value (more or less 30 years). This view of the chemical lifetime as an e-folding time may easily convince that once a mechanism affecting the tracer production OR the tracer sink (in our case, the OH perturbation associated to SG) is suddenly stopped, then several decades are needed to go back to the unperturbed (RCP4.5) lifetime, simply because an increased atmospheric mass of the tracer has to be processed by an amount of OH close to its unperturbed RCP4.5 value. In addition, OH does not instantaneously come back to the original unperturbed value, due to many other changes that have taken place in the atmosphere (SSTs, much longer lived species as $N_2O$ and their indirect impact on $O_3$, etc). But even if OH were instantaneously adjusted to the unperturbed RCP4.5 value, the tracer mass would now start its evolution from a higher value (see Fig. 15). Last, but not least, the stratospheric lifetime of $CH_4$ is much longer than its global ~10 years value, arriving to values even larger than 100 years (above the tropopause). Hope this clarifies.

Page 29: -Line 6: please change to "we have described that an injection of 5-8 Tg of SO2 per year would have effect on large scale. . ."
-Line 6-8: reorganize this sentence, maybe break into two sentences.

We have used these suggestions to rephrase to sentences better.

Page 30:
-Line 1 and Figure 18: "a decrease in tropospheric UV" will be misleading. Figure 14 shows the reduction in only in tropics, and there is an increasing over mid-high latitude.

As before, we have added "tropical" because that is what drives the methane lifetime.

-Please discuss the uncertainty of this study, and what could be improved in future studies.

This has been done in the conclusion. We also modified Figure 18 so as to include the suggestion of the comment posted on ACPD by R. de Richter, and added in the conclusion all possible effects that were not included in our experiments.

Response to the Short Comment posted by R. de Richter

Comment is in blue, author response is in black.

The manuscript acp-2017-593 proposed by D. Visioni et al, is very interesting and deserves publication.

We would like to thank the writer for his comment and for taking the time to read the manuscript. We have tried to address his observations below.

Nonetheless the reader might feel that some important starting hypothesis to their study is missing and should be clearly indicated.

As a matter of fact, as it is written, the manuscript lets us make the assumption that the authors only considered the effects on the newly injected sulphates in the stratosphere by the SRM technology, without taking into consideration the current tropospheric anthropogenic emissions of SO2 and their future evolution during the period in consideration. First, we think that, with the assumption that current anthropogenic sulphur tropo- spheric emissions stay stable during all the period of this study, adding extra-sulphate emissions in the stratosphere would probably increase its global deposition more evenly distributed worldwide than current tropospheric emissions. Under sulphate SRM some wetlands that previously receive low amounts or did not receive tropospheric sul- phates will receive (more) sulphates, and it is known that sulphate in acid rain sup- presses methane emissions from natural freshwater wetlands (Gauci et al, 2008, J. Geophys. Res.), rice paddies, peat lands and other terrestrial landscapes (Oeste and al, 2107, ESD), which are the biggest methane emitters as the authors noted in table 7 of their manuscript; thus CH4 emissions reduction will occur.
Also, it is known that under a global warming (without sulphur SRM), warmer temper- atures and increased rainfall in some regions will increase CH4 emissions. Under the cooling SRM scenarios envisioned by the authors (first column of figure 18 of page 30), the reverse should occur.
Two new columns in figure 18 can be added as follows:
Increase in planetary albedo => surface cooling => lower temperatures => lower CH4 emissions => lower CH4 atmospheric concentration => shorter CH4 lifetime
Increase in planetary albedo => surface cooling => lower rain fall => smaller wetlands area => lower CH4 emissions => lower CH4 atmospheric concentration => shorter CH4 lifetime
We believe the above mentioned assumption (current anthropogenic sulphur tropo- spheric emissions stay stable during all the studied period) should be stated in this manuscript, as:
a) current tropospheric sulphur anthropogenic emissions are and order of magnitude larger than the ones envisioned by the authors for stratospheric SRM;
b) since China's SO2 emissions started decreasing, the current trend is to a global de- crease of tropospheric sulphur anthropogenic emissions (Klimont et al, 2013, Environ. Res. Lett.);
c) estimates of the amounts of sulphur pollution needed to reduce CH4 emissions of the total wetland source have been made (Gauci et al, 2004, PNAS).
Second, the "clathrate gun hypothesis" has been debated by the scientific community as under a warming world, increased emissions from permafrost and/or from methane hydrates destabilisation is a risk. Recent work (Kohnert et al , 2017, Sci. Rep.) sug- gests that a new pathway of CH4 emissions exist and that it may increase if ongoing permafrost thaw continues. Under the cooling SRM scenarios envisioned by the authors the reverse should occur.

One new column in figure 18 page 30 can be added as follows: Increase in plan- etary albedo => surface cooling => lower temperatures => lower CH4 emissions by permafrost => lower CH4 atmospheric concentration => shorter CH4 lifetime.

We agree with the commenter that there are many effects that we have not considered, and considering that we used CCMs for our experiments, we felt it was clear what the limitations were. However, following his suggestion, we have added in the conclusion most of the commenter's remarks regarding other possible side-effects concerning sulfate geoengineering and CH$_4$. We felt, however, that changing Fig. 18 (now Fig. 14 in the revised manuscript) was not the right course of action. We have updated the title, to show what kind of effects we are referring to, and have added in the caption the ones we are not considering.

Third, we agree that the OH radical sink for CH4 is the most important in the tropo- sphere, but it is known than the chlorine radical sink for CH4 is not only important in the stratosphere, but also occurs in the troposphere (Oeste and al, 2107, ESD), where it represents 3-5% of the CH4 removal. Variations in the tropospheric acidity may change the importance of the chlorine sink for methane. With the assumption that cur- rent anthropogenic sulphur tropospheric emissions stay stable during all the period of the author's study, adding extra-sulphate emissions in the stratosphere would probably increase the tropospheric Cl content, and, as the kinetics of the reaction of Cl radical with alkanes (including methane) are an order of magnitude larger than with the OH radical, thus the chlorine radical sink for CH4 will increase.
One new column and a new line in figure 18 page 30 can be added as follows: In- crease in sulphur emissions => increased tropospheric acidity => more HCl increased Cl radical sink for CH4 => more Cl => lower CH4 lifetime
We believe that the authors should add in their manuscript that they made the assumption that this second CH4 sink (the Cl radiacal) is assumed to stay constant in their model.

We would like to better clarify this point with the commenter: our model (ULAQ-CCM) has online an explicit and detailed chlorine-bromine photochemistry; all related species follow the prescribed time evolution by the RCP scenario in use (RCP4.5 in our case). The same also applies to GEOSCCM, by the way. In what is now Table 5, furthermore, regarding sinks and sources of methane, Cl is present as a sink and it would not be correct to state that it stays constant in time.

Response to the Short Comment posted by P.J. Nowack

Comment is in blue, author response is in black.

I think your results are also interesting in connection to air pollution under SRM - could you discuss the possible wider implications briefly in your conclusions? For context, see for example

Xia, L., Nowack, P. J., Tilmes, S., and Robock, A.: Impacts of Stratospheric Sulfate Geoengineering on Tropospheric Ozone, Atmos. Chem. Phys. Discuss., https://doi.org/10.5194/acp-2017-434, accepted for publication. https://www.atmos-chem-phys-discuss.net/acp-2017-434/

Nowack, P. J., Abraham, N. L., Braesicke, P., and Pyle, J. A.: Stratospheric ozone changes under solar geoengineering: implications for UV exposure and air quality, Atmos. Chem. Phys., 16, 4191-4203, https://doi.org/10.5194/acp-16-4191-2016, 2016.

Thank you for your comment. We think that adding some discussion on air pollution could greatly benefit our conclusions. The first paper you mentioned is already cited in regards to ozone depletion (page 22, line 9 in the discussion paper), but will also be included in the conclusions.

We have added a paragraph in the revised manuscript at the end of page 25. It states: *"
[revised manuscript text omitted]